

# Implementation and evaluation of updated photolysis rates in the EMEP MSC-W chemical transport model using Cloud-*J* v7.3e

Willem E. van Caspel[1], David Simpson[1,2], Jan Eiof Jonson[1], Anna M.K. Benedictow[1], Yao Ge[1], Alcide di Sarra[3], Giandomenico Pace[3], Massimo Vieno[4], Hannah L. Walker[4,5,†], and Mathew R. Heal[5]

[1]Norwegian Meteorological Institute, Oslo, Norway
[2]Department of Space, Earth and Environment, Chalmers University of Technology, Gothenburg, Sweden
[3]ENEA Laboratory of Observations And Measurements for the Environment and Climate, Rome, Italy
[4]UK Centre for Ecology & Hydrology, Bush Estate, Penicuik, Edinburgh EH26 0QB, UK
[5]School of Chemistry, University of Edinburgh, Joseph Black Building, David Brewster Road, Edinburgh, EH9 3FJ, UK
[†]Now at Ricardo Energy & Environment, Blythswood Square, Glasgow, UK

**Correspondence:** W.E. van Caspel (willemvc@met.no)

**Abstract.**

The present work describes the implementation of the state of the art Cloud-*J* v7.3 photolysis rate calculation code in the EMEP MSC-W chemical transport model. Cloud-*J* calculates photolysis rates and accounts for cloud and aerosol optical properties at model run-time, replacing the old system based on tabulated values. The performance of Cloud-*J* is evaluated against

5   aerial photolysis rate observations made over the Pacific Ocean, and against surface observations from three measurement sites in Europe. Numerical experiments are performed to investigate the sensitivity of the calculated photolysis rates to the spatial and temporal model resolution, input meteorology model, simulated ozone column, and cloud effect parameterization. These experiments indicate that the calculated photolysis rates are most sensitive to the choice of input meteorology model and cloud effect parameterization, while also showing that surface ozone photolysis rates can vary by up to 20% due to daily variations in

10   total ozone column. Further analysis investigates the impact of Cloud-*J* on the oxidizing capacity of the troposphere, aerosol radiative effect, and surface air quality predictions. Results find that the total tropospheric hydroxyl budget is increased by 26%, while the radiative impact of aerosols is mostly limited to large tropical biomass burning regions. Overall, Cloud-*J* represents a major improvement over the tabulated system, leading to improved model performance for predicting carbon monoxide and daily maximum ozone surface concentrations. The bias is worsened for nitrogen dioxide, however, possibly hinting at model

15   shortcomings elsewhere.





# 1 Introduction

Atmospheric chemistry is driven by a combination of emissions and meteorology, with solar radiation being an essential component, altering the composition and reactivity of the atmosphere through the photolysis of molecules upon absorption of sunlight (Prather et al., 2017). Photolysis reactions also play an important role in air quality, serving as major pathways for the production and loss of atmospheric pollutants such as $O_3$, $NO_x$ ($NO + NO_2$), and volatile organic compounds (VOC) (Mellouki et al., 2015; Sillman, 1999). In the troposphere, the main factors impacting the availability of radiation for photolysis are the solar zenith angle, surface reflections, ultraviolet absorption by stratospheric ozone, and scattering and absorption by cloud and aerosol particles (Real and Sartelet, 2011; Voulgarakis et al., 2009). The representation of these effects is therefore an essential part of any Chemical Transport Model (CTM), which are typically used to simulate the abundance and temporal evolution of atmospheric pollutants.

The CTM developed at the Meteorological Synthesising Centre – West of the European Monitoring and Evaluation Programme (EMEP MSC-W, hereafter "EMEP model") is a three-dimensional Eulerian model, as described in detail by Simpson et al. (2022, 2012) and others (e.g., Ge et al., 2022, 2021; Simpson et al., 2020; Jonson et al., 2018; Stadtler et al., 2018). While the main aim of the model is to provide air quality policy and scientific research support (Jonson et al., 2018; Simpson, 2013), it is also used for operational air quality forecasting (Pommier et al., 2020; Marécal et al., 2015). In its previous configurations, the EMEP model used pre-calculated clear-sky and cloud-sky photolysis rates, often referred to as $J$-values, based upon the work of Jonson et al. (2001). The current work describes the implementation of the Cloud-$J$ v7.3e photolysis rate calculation code, which is now the default scheme in the EMEP model. Cloud-$J$ is a multi-scattering eight-stream radiative transfer model, incorporating aerosol, gas, and cloud radiative properties at model run-time. Cloud-$J$ builds upon the established and computationally efficient Fast-$J$ code (Prather, 2015; Wild et al., 2000), which is used by a wide range of CTMs (e.g., Hall et al., 2018; Marelle et al., 2017; Telford et al., 2013; Søvde et al., 2012; Real and Sartelet, 2011). A key feature of Cloud-$J$ is its flexible handling of cloud scattering effects, offering a choice of eight different numerical schemes at various degrees of computational cost.

In Section 2, an overview of the global EMEP model configuration is provided, including a description of the tabulated and Cloud-$J$ photolysis rate schemes. The implementation and comparison of the two photolysis rate schemes within the standard EMEP EmChem19 chemical mechanism (Bergström et al., 2022) is also described, using the boxChem testing tool (Simpson et al., 2020). In Section 3, the simulated photolysis rates are compared against aerial observations over the Pacific Ocean made during the Atmospheric Tomography Mission flight campaign (ATom-1, Wofsy et al., 2021), while Section 4 includes comparisons against surface measurements from the Chemistry-Aerosol Mediterranean Experiment (ChArMEx 2013, Mallet et al., 2016), CYprus PHotochemical EXperiment (CYPHEX 2014, Meusel et al., 2016), and from Chilbolton, England. In the comparisons against aerial and surface observations, the sensitivity of the calculated photolysis rates to different model configuration is also investigated. These include the model resolution, choice of photolysis and cloud effect scheme, input meteorology model, and simulated $O_3$ column. In Section 5, the impact of the updated photolysis rates on the oxidizing capacity of the troposphere is investigated, along with the impact of the newly introduced aerosol radiative effect. In this



50    section, comparisons against surface observations of $O_3$, $NO_2$, and CO concentrations from the EBAS database (Tørseth et al., 2012) are also included. The results are summarized and concluded in Section 6.



## 2 Model Description

### 2.1 Model Configuration

As noted above, previous versions of the EMEP model made use of tabulated photolysis rate calculations. In this study, we
use an updated version of the EMEP model, v4.47, which includes Cloud-$J$, and enables side-by-side comparisons of the
two methods on a global scale. To this end, the model incorporates global meteorological fields from the ECMWF Integrated
Forecasting System (IFS) cycle 40r1 model (ECMWF, 2014) on a 0.5° x 0.5° latitude-longitude grid. The model time-step
is grid-size dependent, and is 20 minutes for the global simulations presented in the current work. The meteorological fields
include specific humidity, horizontal and vertical winds, potential temperature, grid-fraction cloud cover, and cloud ice and
cloud liquid specific water content. The EMEP model is run with 19 vertical hybrid pressure-sigma levels extending between
the surface and 100 hPa, where the lowest layer is approximately 90 meters thick. However, output surface concentrations are
adjusted to an equivalent height of 3 meters to account for gradients induced by dry-deposition (c.f. Simpson et al., 2012).
Also as part of the model upgrade to v4.47, 3-hourly IFS $O_3$ concentrations are now specified at the top boundary and a fixed
global mean background concentration of 500 ppb is used for $H_2$ gas. For $CH_4$ global mean background concentrations follow
the yearly mean values reported as part of CMIP6 AR6 (Masson-Delmotte et al., 2021), having a value of 1858 ppb for the
simulation for the year 2018 discussed in Section 5.

A global annual emission inventory based on the ECLIPSEv6b (Evaluating the CLimate and Air Quality ImPacts of Short-
livEd Pollutant version v6b) data set is used, which contains annual gridded emissions of $SO_2$, $NO_x$, $NH_3$, CO, $CH_4$, NMVOCs
(Non-Methane VOC), primary fine Particulate Matter ($PM_{2.5}$), and primary coarse PM ($PM_{co}$), including the contributions
of international shipping. Forest fire emissions are specified using the FINNv2.5 (Fire INventory from NCAR version 2.5,
Wiedinmyer et al. 2023) data set, which succeeds the FINNv1.5 data set described in Wiedinmyer et al. (2011). The FINN2.5
dataset includes daily emissions for a number of species, including NOx, $SO_2$, NMVOC, $PM_{2.5}$, and organic and black carbon
aerosol, which in the model are distributed evenly within the model layers below 800 hPa.

### 2.2 Tabulated Photolysis Rates

The EMEP model version when driven by tabulated photolysis rates (hereafter "EMEP-TB") uses seasonal look-up tables of
clear and cloudy-sky (all-sky) photolysis rates, following Jonson et al. (2001). These tables were calculated as a function of
SZA (0-90°) using the two-stream PHODIS routine described in Kylling et al. (1998, 1995), incorporating cross-sections and
quantum yield data from DeMore et al. (1997). The cloudy-sky photolysis rates were calculated for two predefined cloud fields
at 55°N. Namely, thin and light clouds having a water content of 0.3 g cm$^{-3}$ and mean droplet radius of 6 $\mu$m, and thick and
dense clouds having a water content of 0.7 g cm$^{-3}$ and mean droplet radius of 10 $\mu$m. For overhead stratospheric $O_3$, results
from the 2-D global model described in Stordal et al. (1985) were used, scaled by observed total $O_3$ column (TOC) observations
from Dütsch (1974).

The cloudy-sky photolysis rates are tabulated as a function of solar zenith angle (SZA) and altitude. However, the clear-sky
photolysis rates are also tabulated as a function of latitude, between 30°-90°N in 10° bins. Between 30°S and 30°N photolysis





rates from the 30°-40° latitude bin are used, while between 30°-90°S they mirror those from 30°-90°N. We note that the range
      of latitudes for the clear-sky photolysis rates reflects the geographical extent of the traditional EMEP domain. To calculate
      photolysis rates at model run-time, the cloud cover type in each vertical column, or Independent Column Atmosphere (ICA),
      is first determined by calculating the cloud thickness. If clouds are present, clouds less than 1.5 km between the cloud top and
      cloud base are classified as thin and light, while clouds with a greater vertical extent are classified as thick and dense. The

maximum cloud cover fraction within the ICA is then used as a weighting term to linearly interpolate between the clear-sky
      and cloudy-sky photolysis rates. Aerosol scattering and absorption effects are not considered in the tabulated scheme.

## 2.3   Cloud-$J$ v7.3e

The Cloud-$J$ v7.3e code builds on version v7.3d described in Prather (2015), incorporating only a few minor numerical bug
fixes. Cloud-$J$ is based upon its Fast-$J$ predecessors, which has a history of development and comparison to observation

spanning over two decades (e.g., Hall et al., 2018; Sukhodolov et al., 2016; Barnard et al., 2004). Its highly optimized eight-
      stream radiative transfer scheme employs 18 wavelength bins for wavelengths between 177 nm and 778 nm (Neu et al.,
      2007), spanning the wavelengths relevant to tropospheric and stratospheric chemistry. However, since EMEP is a tropospheric
      model, wavelengths relevant to the stratosphere (<196 nm) are not included, reducing the number of wavelength bins to 12.
      While Cloud-$J$ solves the radiative transfer equations over a small number of wavelength bins, the accuracy of the calculated

photolysis rates is nevertheless maintained within a few percentage points (Wild et al., 2000). The EMEP model running with
      Cloud-$J$ is hereafter referred to as "EMEP-CJ".

      Cloud-$J$ incorporates quantum-yield and cross-sectional data for each of the photolysed species and for each of the wave-
      lengths bins, based on the molecular data recommendations from Sander et al. (2011) and Atkinson et al. (2008) by default
      (as will be discussed in more detail in Section 2.4.1). These binned molecular data are used in the propagation of the top of

the atmosphere solar irradiance spectrum through each ICA, taking into account scattering and absorption by gases, aerosols,
      and clouds, in addition to surface reflections. In EMEP, the surface albedo is calculated using a mosaic-approach, where a
      grid-box weighted average is calculated based on the different land-types present within each surface grid-box. For scattering
      and absorption by clouds, the liquid and ice cloud water content and the cloud cover fraction fields from the input meteorology
      are used. The cloud optical properties are calculated using the standard formulae provided with Cloud-$J$, which calculate the

cloud liquid and ice effective particle radii as empirical functions of pressure and ice loading (g m$^{-3}$), respectively. For the
      radiative impact of clouds, the G6/.33 MAX-COR model for cloud overlap with Averaged Quadrature Column Atmospheres
      (MAX-COR AvQCA) scheme is used by default, as recommended by Prather (2015). This scheme employs a cloud correlation
      factor of 0.33 with 6 maximally overlapping correlated (MAX-COR) groups, requiring on average 2.8 calls to the photolysis
      rate calculation scheme per ICA. As an alternative, the Briegleb averaging method (Briegleb, 1992) is discussed in Sections 3

and 5.3, requiring only a single call per ICA.





### 2.3.1 Aerosol Scattering and Absorption

Aerosols impact photolysis rates through the scattering and absorption of solar radiation, typically leading to a decrease in photolysis rates near the surface (Gao et al., 2020; Xing et al., 2017; Gerasopoulos et al., 2012; Casasanta et al., 2011; Tie et al., 2005). This effect, referred to as the aerosol direct effect, is generally largest in the continental summertime lower

troposphere, where a majority of the effect can be attributed to the presence of dust and biomass burning aerosol (Bian et al., 2003). Biomass burning aerosol can reduce surface photolysis rates by as much as a factor of two (Martin et al., 2003), affecting both regional $O_3$ and OH abundances. Over the oceans, the aerosol direct effect is dominated by sea salt aerosol (Murphy et al., 1998).

In EMEP-CJ, the aerosol direct effect is calculated using tabulated scattering phase function and single-scattering albedo

values from the University of Michigan (UMich) data set, which is distributed along with the Cloud-$J$ code. The UMich optical properties are used to calculate the radiative impact of sea-salt, dust, and biomass burning aerosol, using their instantaneous abundances as simulated by the EMEP model. The UMich optical properties are tabulated as a function of ambient Relative Humidity (RH), between 0-99% in 5% intervals, which in the EMEP model is calculated from the input meteorology. The mass of the fine and coarse mode log-normal distributions used by the EMEP model is distributed over the respective UMich radius

bins using the log-normal mass-fraction formula (Seinfeld and Pandis, 2016).

Modeled sea salt is generated as a function of the surface wind speed, as discussed in detail in Tsyro et al. (2011). Since sea salt is highly hygroscopic (Zieger et al., 2017), the parameters of the log-normal distributions (mass median diameter and geometric standard deviation) are calculated as a function of RH using the empirical functions of Gerber (1985). For biomass burning, the aerosol optical properties depend on the black carbon mass-fraction of the total biomass burning aerosol (e.g.,

Bond and Bergstrom, 2006), which in the EMEP model is calculated from the instantaneous abundance of the forest fire species from the FINNv2.5 data set, as discussed in Section 2.1. Dust aerosol is generated as a function of land-cover type and wind speed (Simpson et al., 2012). The net impact of the aerosol direct effect on the EMEP-CJ simulation results is discussed in more detail in Section 5.2.

### 2.3.2 Stratospheric $O_3$

The photolysis of $O_3$ for wavelengths below 320 nm is an important loss mechanism for tropospheric $O_3$, while the subsequent reaction of $O(^1D)$ with water vapour is the main source of tropospheric OH (Fuglestved et al., 1994). The photolysis of tropospheric $O_3$ is highly sensitive to the overhead stratospheric $O_3$, which acts as an absorber of the relevant wavelengths (Casasanta et al., 2011). In EMEP-CJ, the overhead stratospheric $O_3$ is specified using global stratospheric measurements from the MErged GRIdded Dataset of Ozone Profiles (MEGRIDOP). As described in detail in Sofieva et al. (2021), the MEGRIDOP

data set is based on merged observations from the GOMOS, MIPAS, OSIRIS and SCIAMACHY limb-scanning satellites. These observations are combined to construct monthly mean gridded vertical profiles of the atmospheric temperature and mole concentration of $O_3$ in air, extending roughly between the tropopause and stratopause (ca. 10-50 km altitude). The horizontal grid is spaced $10° \times 20°$ in latitude-longitude, while the vertical resolution is 2–4 km.



For the Cloud-$J$ photolysis rate calculations, the MEGRIDOP satellite observations are appended to the EMEP model levels,
resulting in 16 additional levels in the standard 100 hPa model top configuration. The 16 extra levels correspond to a factor of
two vertical sub-sampling of the satellite measurements, which is found to have a negligible impact on the simulated $J$-values.
Using the uncertainty estimates on the retrieved $O_3$ profiles, the total measurement uncertainty is estimated to be less than
3 Dobson Units for the stratospheric $O_3$ column, or less than 1-2% of the total. The MEGRIDOP observations are available
between the years 2002 and 2021, while a climatology based on these years is used for other years, including future years.
Diagnostic analysis where the year-to-year dataset of stratospheric $O_3$ is replaced by the climatology, find that inter-annual
variability in stratospheric $O_3$ leads to surface ozone variations of 0.5-1.0 ppb, predominantly during spring and early summer.

## 2.4 BoxChem

BoxChem is a zero-dimensional boundary layer chemistry box model, serving as a testing tool for chemical reaction mecha-
nisms in the GenChem and EMEP modeling systems (Simpson et al., 2020). The aim of using boxChem in the current work
is to implementat and test the impact of the different photolysis rate schemes on the default EMEP EmChem19 chemical
mechanism.

For the Cloud-$J$ implementation, the stand-alone climatological Cloud-$J$ reference program is used. In this configuration,
Cloud-$J$ reads in climatological background atmospheric chemical and meteorological fields, and excludes aerosol effects. For
the purpose of the boxChem simulations, only clear-sky conditions are used for both the Cloud-$J$ and tabulated schemes.

### 2.4.1 EmChem19

The default EmChem19 chemical mechanism of the EMEP model succeeds the EmChem16 (EMEP status report, 2017) and
EmChem09 (Simpson et al., 2012) mechanisms. A key feature of EmChem19 is its aim to balance computational complexity
with realism by using a simplified set of surrogate VOCs, as discussed in detail in Bergström et al. (2022). In the context
of the current work, notable features of EmChem19 are its 16 primary photolysis reactions, here referring to those reactions
which are specific to a certain reactant. Secondary reactions are taken to be those reactions where photolysis rates from one
of the primary reactions is used as a (scaled) surrogate, and are not discussed in further detail here. The first column of
Table 1 gives an overview of the primary photolysis reactions (R1-R16) in EmChem19. The second column gives additional
information on the temperature and pressure dependence of the photolysis rates calculated by Cloud-$J$. While the focus in
the current work is on the default EmChem19 mechanism, a brief overview of the Cloud-$J$ implementation in the Common
Representative Intermediates (CRI) Version 2 Reduced variant 5 (CRI v2-R5, Watson et al., 2008) chemical mechanism is
included in Appendix A1.

The molecular data distributed along with the standard Cloud-$J$ code is used for the photolysis rate calculations for R1-R15,
which is based on quantum-yield and cross-sectional data recommended by Sander et al. (2011) for R1-R13. For R14 and
R15, the molecular data is based on the recommendations from Atkinson et al. (2008). However, Cloud-$J$ is used to extend
EmChem19 by explicitly specifying the three glyoxal (CHOCHO) photolysis channels R17a-R17c. In addition, the Cloud-$J$
data is extended for the photolysis of biacetyl (R16). For this, cross-sectional data between 206 and 496 nm at 1 bar and





298 K from Horowitz et al. (2001) are used, along with an effective quantum yield of $\phi = 0.158$ between 290-470 nm Plum et al. (1983). The quantum yield is assumed to be zero for wavelengths outside of the 290-470 nm range, as >99% of the photodissociation occurs in the 340-470 nm absorption band (Plum et al., 1983).

To test the sensitivity of EmChem19 to the updated photolysis rates, boxChem simulations are run for the 1st of July at 45°N and 15° East, corresponding roughly to summertime central Europe. Following the boxChem setup of Bergström et al. (2022, Table 3.1), background mixing ratios for $CH_4$, CO, and $H_2$ are set to 1800 ppb, 120 ppb, and 400 ppb, respectively. The atmospheric temperature is set to 298.15 K, with a RH of 66.5%. A mixing height of 1 km is assumed for the vertical dispersion of emissions. Anthropogenic $NO_x$ and VOC emissions are kept constant at a rate of 18.3 kg km$^{-2}$ day$^{-1}$ and 15.4

kg km$^{-2}$ day$^{-1}$, respectively, similar to those used by Jenkin et al. (2017, 2008) and Watson et al. (2008). These emissions lead to a simulated daily averaged $NO_x$ mixing ratio of approximately 2.5 ppb, broadly matching observed background surface concentrations over continental Europe. The model is run for a period of three days, after which time daily mean concentrations of the photolysed species are calculated.

     The third and fourth columns in Table 1 show the calculated tabulated and Cloud-$J$ surface photolysis rates at 12:00 hr

local time, respectively, as an indication of the photolysis strength. The fifth column ($\delta C$) shows the change in daily mean concentration of the photolysed species when only the photolysis rate specific to the reaction in each row is changed to Cloud-$J$. The sixth column ($\delta C_{tot}$) shows the change in concentration when all photolysis rates are changed to those from Cloud-$J$. The difference between $\delta C$ and $\delta C_{tot}$ thus illustrate that, while the photolysis strength of a single reaction may be higher or lower in Cloud-$J$, the net impact of the Cloud-$J$ rates may yield a different net change due to the interlinked chemistry. This

is illustrated by R6, where direct photolysis is reduced whereas the net concentration change is negative.

     Table 1 illustrates that the net change in the daily mean concentrations due to the photolysis rate update is generally on the order of $\pm$ 5-10%. Exceptions occur for R9 and R13, which see a net change of 29.4 and -18.9%, respectively, owing largely to the changes in their direct photolysis. We note that the standard Cloud-$J$ code treats the ratios of NO and $NO_2$ production by R8 as 0.114 and 0.886, respectively, whereas this ratio is 0.127 and 0.873 in EmChem19. However, using boxChem the different

ratios are found to impact simulated $NO_x$ and $O_3$ concentrations by no more than 0.1%, such that the default EmChem19 ratio is kept by default. In addition, Table 1 illustrates that the daily mean concentration for biacetyl (R16) is reduced by 8.31%, even though the noon-time Cloud-$J$ rate is only 0.68% greater than the tabulated value. However, further analysis finds that the diurnal cycle in Cloud-$J$ is broader than in the tabulated scheme, with reaction rates being higher by 5.9% on average over the course of the day.





**Table 1.** Overview of the primary photolysis reactions in EmChem19. Reactions R17a–R17c are newly introduced with Cloud-$J$, and together replace R13.

| Reaction | Cloud-J Notes | Tab[a] (s$^{-1}$) | Cloud-J[a] (s$^{-1}$) | $\delta C$ (%)[b] | $\delta C_{t,tot}$ (%)[c] |
|---|---|---|---|---|---|
| 1. $O_3 \longrightarrow O_2 + O(^1D)$ | T-dependence between 218-298 K | $3.89 \times 10^{-5}$ | $2.91 \times 10^{-5}$ | +2.94 | +6.59 |
| 2. $O_3 \longrightarrow O_2 + O(^3P)$ | T-dependence between 218-298 K | $4.49 \times 10^{-4}$ | $4.94 \times 10^{-4}$ | 0.0 | +6.59 |
| 3. $NO_2 \longrightarrow NO + O(^3P)$ | T-dependence between 200-300 K | $8.15 \times 10^{-3}$ | $9.46 \times 10^{-3}$ | -3.28 | -1.56 |
| 4. $H_2CO \longrightarrow HCO + H$ | T-dependence between 223-298 K | $2.84 \times 10^{-5}$ | $3.18 \times 10^{-5}$ | -1.48 | -4.98 |
| 5. $H_2CO \longrightarrow CO + H_2$ | T-dependence between 223-298 K | $4.30 \times 10^{-5}$ | $4.93 \times 10^{-5}$ | -1.53 | -4.98 |
| 6. $H_2O_2 \longrightarrow 2\,OH$ | T-dependence between 200-300 K | $6.96 \times 10^{-6}$ | $6.85 \times 10^{-6}$ | +0.69 | -8.86 |
| 7. $CH_3OOH \longrightarrow CH_3O + OH$ | | $4.88 \times 10^{-6}$ | $5.25 \times 10^{-6}$ | -0.13 | -12.33 |
| 8. $NO_3 \longrightarrow 0.127\,NO + 0.127\,O_2 + 0.873\,NO_2 + 0.873\,O$ | T-dependence between 190-298 K | 0.21 | 0.22 | -0.01 | +8.11 |
| 9. $HNO_2 \longrightarrow NO + OH$ | | $1.80 \times 10^{-3}$ | $1.52 \times 10^{-3}$ | +33.76 | +29.37 |
| 10. $HNO_3 \longrightarrow NO_2 + OH$ | T-dependence between 200-300 K | $1.38 \times 10^{-7}$ | $5.88 \times 10^{-7}$ | -0.15 | -3.15 |
| 11. $HNO_4 \longrightarrow NO_2 + HO_2$ | | $1.35 \times 10^{-6}$ | $1.82 \times 10^{-5}$ | -0.02 | -5.53 |
| 12. $CH_3COCHO \longrightarrow CH_3CO + CO$ | P-dependence between 177-999 hPa | $1.55 \times 10^{-4}$ | $1.80 \times 10^{-4}$ | -4.86 | -2.04 |
| 13. $CHOCHO \longrightarrow 1.9\,CO + 0.1\,HCHO + 0.5\,HO_2$ | Calculated as the sum of R17a-R17c | $6.54 \times 10^{-5}$ | $1.19^d \times 10^{-4}$ | -18.56 | -18.92 |
| 14$^e$. $MEK \longrightarrow CH_3CO_3 + C_2H_5O_2$ | P-dependence between 177-999 hPa | $1.03 \times 10^{-5}$ | $3.63 \times 10^{-6}$ | +14.9 | +13.86 |
| 15. $CH_3CHO \longrightarrow CH_3O_2 + HO_2 + CO$ | P-dependence between 177-999 hPa | $1.01 \times 10^{-5}$ | $3.48 \times 10^{-6}$ | +3.54 | -0.55 |
| 16. $CH_3C(O)C(O)CH_3 \longrightarrow 2\,CH_3CO_3$ | P-dependence between 177-999 hPa | $2.94 \times 10^{-4}$ | $2.96 \times 10^{-4}$ | -8.31 | -11.05 |
| 17a. $CHOCHO \longrightarrow 2\,HCO$ | P-dependence between 177-999 hPa | | $7.45 \times 10^{-5}$ | | |
| 17b. $CHOCHO \longrightarrow H_2 + 2\,CO$ | P-dependence between 177-999 hPa | | $1.59 \times 10^{-5}$ | | |
| 17c. $CHOCHO \longrightarrow CH_2O + CO$ | P-dependence between 177-999 hPa | | $2.88 \times 10^{-5}$ | | |

a) Surface clear-sky photolysis rates on DOY 182 at 12:00 hr local time at 45° N and 15° E.
b) Change in daily mean concentration (%) after 3 days of integration when only the reaction rate specific to each row is changed to Cloud-$J$.
c) Change in daily mean concentration (%) after 3 days of integration when Cloud-$J$ photolysis rates are used for all photolysis reactions.
d) In EmChem19 the product of glyoxal (CHOCHO) photolysis is based on a fixed relative strength assumption of the R17a-c channels. Here the net photolysis rate from Cloud-$J$ is taken as the sum of R17a-c.
e) Methyl Ethyl Ketone (MEK = $CH_3C(O)CH_2CH_3$) assumes $\alpha = 0$ for the $CH_3 + C_2H_5CO$ photolysis product channel described in Zborowska et al. (2021).





**Table 2.** Photolysis and cloud field model configurations for the ATom-1 data comparison.

| Short name | Long name | Cloud data[a] and date | $J$-value and cloud fraction (CF) treatment |
|---|---|---|---|
| CJ | EMEP-CJ | IFS 3h CF + WP data ($0.5° \times 0.5°$) 16 August 2016 | Cloud-$J$ v7.3e, liquid and ice cloud optical properties per Cloud-$J$, MAX-COR AvQCA cloud effect scheme |
| TB | EMEP-TB | IFS 3h CF data ($0.5° \times 0.5°$) 16 August 2016 | Tabulated photolysis rates |
| CJ15 | EMEP-CJ for the 16th of August 2015 | IFS 3h CF + WP data ($0.5° \times 0.5°$) 16 August 2015 | As CJ |
| CJVL | EMEP-CJ with 36 vertical levels | As CJ | As CJ |
| CJB | EMEP-CJ with Briegleb averaging | As CJ | Cloud-J v7.3e, liquid and ice cloud OD per Cloud-$J$, Briegleb averaging[b] |
| CJW | EMEP-CJ with 3-hourly WRF input meteorology | WRF 3h CF + WP data ($0.5° \times 0.5°$) 16 August 2016 | As CJ |
| CJWH | EMEP-CJ with hourly WRF input meteorology | WRF 1h CF + WP data ($0.5° \times 0.5°$) 16 August 2016 | As CJ |

a) Includes cloud fraction (CF) and cloud ice and liquid water path (WP).

b) Approximates cloud Optical Depth (OD) in single-column atmosphere grid cell as OD(in-cell)=OD(in-cloud)$\times$ CF$^{3/2}$ (Briegleb, 1992).

## 3 ATom-1 Aircraft Campaign

In this section, simulated photolysis rates are compared against observations from the first NASA Atmospheric Tomography (ATom-1) aircraft mission over the Pacific Ocean (Wofsy et al., 2021). During this mission, $J$-O($^1$D) (R2) and $J$-NO$_2$ (R3) were calculated using the Charged-coupled Actinic Flux Spectroradiometer (CAFS) instrument on board the NASA DC-8 research aircraft (Travis et al., 2020). The CAFS instrument measured 4-$\pi$ steradian actinic flux density spectra from 280 to 650 nm, with a sampling resolution of three seconds. The analysis discussed here follows that of Hall et al. (2018), who describe the observational data set in detail, along with the methodology to compare the data against photolysis rate calculations from global CTMs. In the current work, comparisons are made against seven diagnostic EMEP simulations, designed to test the sensitivity of the calculated photolysis rates to different model configurations. An overview of the diagnostic model configurations discussed in this section is given in Table 2.



The ATom-1 deployment consisted of 10 flights between the 29th of July to the 23rd of August 2016, occurring mostly during day time. As described in Hall et al. (2018), the flight data were used to construct a climatological statistic of the observed photolysis rates and cloud-effect over the tropical ($20°$S-$20°$N,160-240°E) and North ($20$-$50°$N,170-225°E) Pacific blocks for $\cos(\text{SZA}) > 0.8$. Here the cloud effect refers to the general tendency of clouds to increase photolysis rates above the cloud layer due to scattering, while diminishing them below the cloud layer due to shadowing. While the CAFS instrument measured all-sky photolysis rates during flight time, a conjugate artificially cleared dataset of clear-sky photolysis rates was constructed using the Tropospheric Ultraviolet and Visible (TUV) radiative transfer model.

## 3.1 Vertical Profiles

Following Hall et al. (2018), global simulations spanning 24 hours on a day in mid-August are used to infer the statistical properties of the simulated $J$-O($^1$D) and $J$-NO$_2$ values. To match the CAFS observations, only model data for where $\cos(\text{SZA}) > 0.8$ is used, for the geographical areas spanning the Tropical and North Pacific blocks. Simulations are performed for both all-sky and clear-sky conditions, where the clear-sky simulations also exclude the effect of aerosols. However, consistent with the findings of Hall et al. (2018), diagnostic simulations find aerosols to have a negligibly small impact on the calculated photolysis rates over the Pacific ocean. While Hall et al. (2018) compare CAFS against model data for simulations anywhere between 15-17 August and the years 2013 to 2017, the EMEP simulations center on the 16th of August 2016.

The baseline EMEP-CJ simulation (CJ, in short) uses the model configuration outlined in the Section 2.1. The CJB experiment uses the Briegleb averaging rather than MAX-COR AvQCA cloud effect scheme, to investigate the impact of using a less numerically demanding cloud effect scheme. The Briegleb scheme modifies the in-cloud optical depth by a factor of $f^{3/2}$, where $f$ is the cloud cover fraction in each grid-box (Briegleb, 1992), to calculate the grid-box average cloud optical depth. An EMEP-CJ simulation for the 16th of August 2015 (CJ15) is included to investigate the impact of year to year meteorological variability on the constructed photolysis rate statistics. In the CJVL experiment, all IFS vertical levels available from the input meteorology files are used, amounting to a two-fold increase over the standard number levels, and a corresponding two-fold increase in the vertical resolution of the cloud field. The EMEP-TB model is included (TB) to provide reference to the tabulated scheme. In addition, EMEP-CJ simulations using input meteorology from the Weather Research and Forecast model (WRF) version 4.4.2 are included (CJW), which has been used to drive the EMEP model in a number of studies (e.g., Ge et al., 2022; Langford et al., 2022; van der Swaluw et al., 2021). The CJW experiment employs the same grid and temporal resolution as those based on the IFS meteorology, whereas the CJWH experiment uses hourly WRF input files instead. The latter experiment investigates the impact of the temporal resolution, noting that hourly data are not available from the IFS input meteorology for the EMEP model. In the current work, the WRF winds and temperatures are nudged to ERA5 reanalysis fields (Hersbach et al., 2020). While the ERA5 model is itself an IFS-based system, the WRF model nevertheless outputs a broad range of other resolved parameters, while also employing its own microphysics scheme (WASM 5-class scheme with ice and supercooled water and snow melt, Hong et al., 2004). To isolate the impact on the photolysis rate calculations, all of the above diagnostic simulations are initialized starting from EMEP-CJ on the 15th of August 2016.





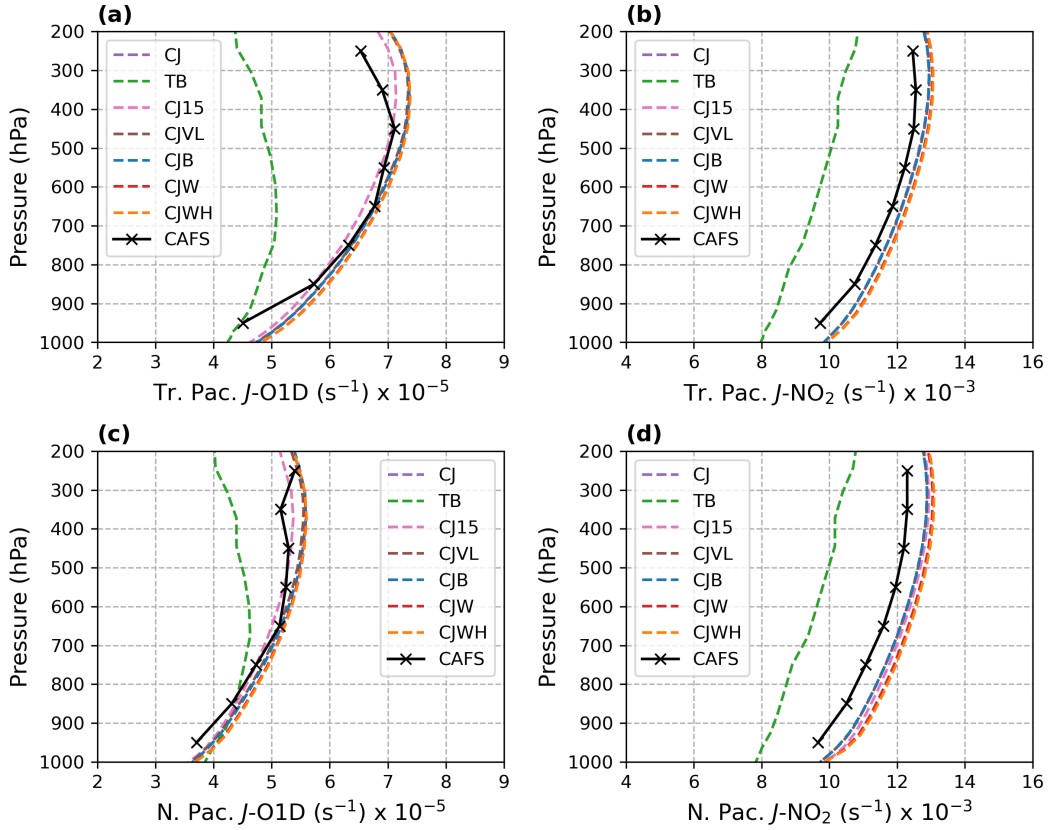

**Figure 1.** Vertical profiles of clear-sky $J$-O($^1$D) and $J$-NO$_2$ over the tropical and North pacific blocks. The CAFS values are directly measured in ATom-1, while the models are sampled over 24 hours on a mid-August day for $\cos(\text{SZA}) > 0.8$. Legend labels are explained in the text. Note that the x-axis does not start at zero.

Fig. 1 shows the model and CAFS comparison for the clear-sky photolysis rates. Focusing first on the Cloud-$J$ model configurations, both the simulated and observed photolysis rates maximize towards the tropopause, where incoming solar

radiation is at its strongest. The calculated clear-sky rates are further nearly identical between the different Cloud-$J$ model configurations, with the exception of the slight deviations present in the temperature-dependent $J$-O($^1$D) rates in the CJ15 experiment. All Cloud-$J$ derived photolysis rates are nevertheless within 15% of the CAFS observations at all altitudes. The TB simulation is markedly different, however, underestimating $J$-O($^1$D) and $J$-NO$_2$ over both the tropical and Northern pacific basins by as much as 30-35% in the middle and upper troposphere, with the difference being greatest in the tropics.

Fig. 2 shows the CAFS all-sky photolysis rate comparisons. While the different IFS-based Cloud-$J$ simulations show relatively little inter-model variations, the comparatively largest differences occur for 1) CJB for $J$-NO$_2$ in both Pacific blocks, 2) CJ15 for $J$-O($^1$D) in both blocks, and 3) CJVL for $J$-NO$_2$ in the Northern pacific block. Both the results for CJVL and CJB are consistent with the Northern pacific being generally more cloudy, and thus experiencing a larger impact of the representation

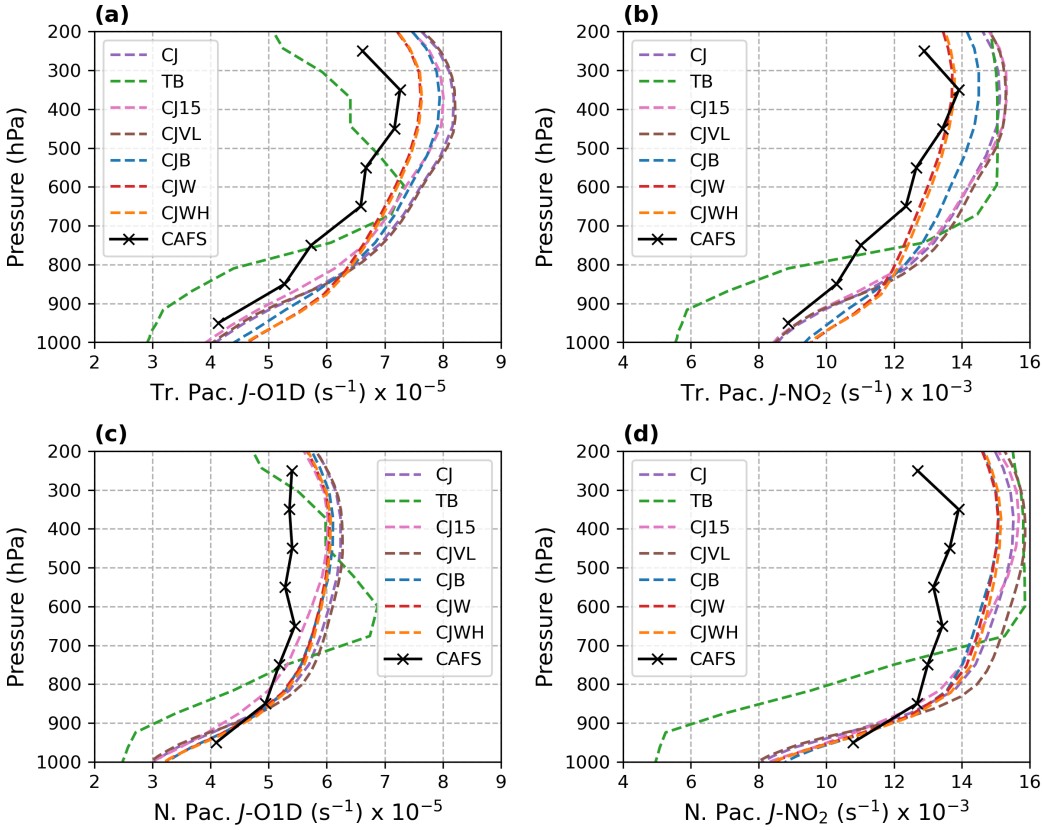

**Figure 2.** Vertical profiles of all-sky $J$-O($^1$D) and $J$-NO$_2$ over the tropical and North pacific blocks. The CAFS values are directly measured in ATom-1, while the models are sampled over 24 hours on a mid-August day for $\cos(\text{SZA}) > 0.8$. Legend labels are explained in the text. Note that the x-axis does not start at zero.

of the cloud effect. The largest difference between the Cloud-$J$ simulations occurs for the WRF-based simulations, which gen-
erally show lower (higher) photolysis rates above (below) $\sim$800 hPa compared to the IFS-based results. The CJW and CJWH results are nearly identical, indicating that the temporal resolution makes a comparatively small impact. The largest contrast occurs for the TB simulation, whose $J$-values below $\sim$800 hPa are considerably smaller than observation, and much greater than observation for $J$-O($^1$D) in the Northern pacific block between 500-700 hPa. For $J$-NO$_2$, the TB values are closer to those of the other models, but only for altitudes above 700 hPa.

To highlight the impact of the cloud-effect, Fig. 3 shows the ratio of the clear-sky to all-sky photolysis rates. In the EMEP-CJ configurations with IFS meteorology, clouds enhance $J$-values by approximately 10-20% in the upper and middle troposphere, while diminishing them by 10-20% closer to the surface. Their respective vertical profiles show general agreement with those derived from CAFS, with the exception of the above-cloud enhancement being stronger in the tropical Pacific. For the latter, the CAFS $J$-values are reduced up to an altitude of 500-700 hPa, whereas the simulated rates are reduced only up to altitudes

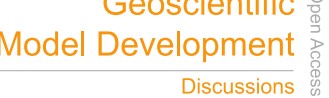

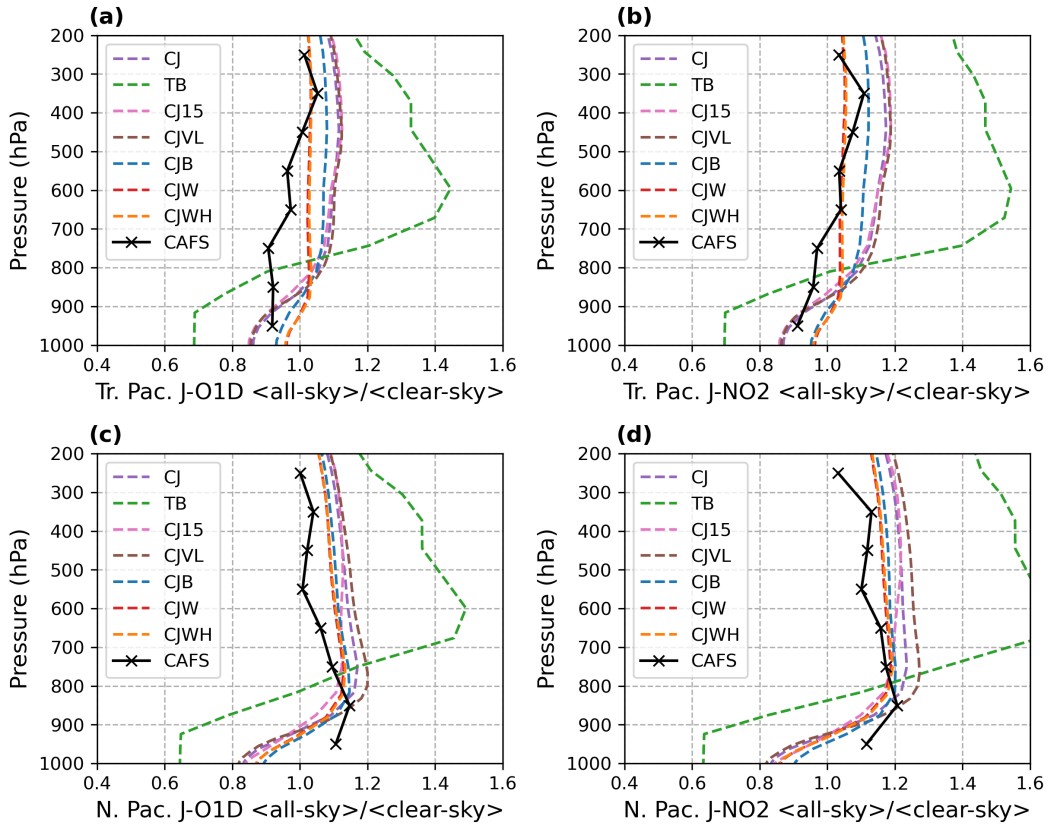

**Figure 3.** Vertical profiles of the ratio of clear-sky to all-sky $J$-O($^1$D) and $J$-NO$_2$ over the tropical and North pacific blocks. The ratios are calculated based on the data shown in Fig. 1 and 2. Legend labels are explained in the text.

below ~950 hPa. This is in line with the general result that the cloud effect is stronger in the IFS-based Cloud-$J$ simulations than observation, with the CAFS ratios being closer to 1 throughout much the atmospheric column. Another notable difference is that the sign of the cloud effect is opposite between the simulations and CAFS for the Northern Pacific below 900 hPa. Since a below-cloud enhancement has no physical basis in the current models, this hints at possibly missing modeled cloud field types (Hall et al., 2018). The cloud effect in the TB simulation stands out as being much stronger than in CAFS or any of the other simulations, both for $J$-O($^1$D) and $J$-NO$_2$.

## 3.2 Cloud Effect Statistics

Another point of view for the comparison of model and observation, is to look at the properties of the distribution of the cloud effect. To this end, the statistics of how often and by how much clouds either increase or diminish the clear-sky $J$-O($^1$D) and $J$-NO$_2$ values are calculated for three different vertical regions. The latter are chosen to be representative of the below-cloud, in-cloud and above-cloud layers, spanning the surface to 900 hPa, 900-300 hPa, and 300-100 hPa, respectively. As the



Northern Pacific is in general more cloudy than the Tropical Pacific, only the results for the Northern Pacific are discussed in the following. These results are, however, also representative for the tropical Pacific, which is included for reference in Fig. S1 of the supplementary material.

Following the approach of Fig. 5 from Hall et al. (2018), the cloud statistics are summarized in Fig. 4 for the CAFS observations and for the different model configurations. In this figure, each of the horizontal bars has a length of 100%, while the length of the left-most thick segment marks the percentage of measurements for which photolysis rates are diminished by clouds. Similarly, the length of the right-most thick line marks the number of percentage points for which clouds enhance the photolysis rates. The length of the middle thin line segment marks the number of percentage points where photolysis rates are neither enhanced or diminished by more than 2.5%. The markers (crosses) placed within the thick line segment denote the average decrease (increase) of the photolysis rates, for those times where the rates are decreased (increased) by more than 2.5%. Crosses are only shown when at least 2% of the photolysis rates are either enhanced or diminished by more than 2.5%. For example, clouds decrease photolysis rates by more than 2.5% roughly 20% of the time for the CAFS $J$-O($^1$D) measurements between 100-300 hPa shown in Fig. 4a, causing a decrease of around 5% on average.

Fig. 4 confirms the picture that the IFS-based EMEP-CJ model configurations generally overpredict the occurrence of enhanced (diminished) photolysis rate in the above (below) cloud layer, although this effect is less pronounced for the CJB simulation. The average enhancement factors in the above-cloud layer nevertheless agree well with observation. In general, the WRF-based simulation shows less above-cloud enhancement and less below-cloud dimming than the IFS-based Cloud-$J$ simulations, while its average enhancement and dimming are stronger. For the in-cloud statistics, all models overestimate the magnitude of the dimming while underestimating its occurrence, while the WRF-based simulations also overestimate the average magnitude of the enhancement. Also here, the CJW and CJWH simulations show nearly identical results. Overall, for the Cloud-$J$ based simulations, the impact of the vertical resolution and meteorological year is small, while the impact of using WRF meteorology over IFS is considerably larger. Furthermore, for the below-cloud layer the impact of using Briegleb averaging is similar to that of using WRF meteorology. We note that the over-occurrence of above-cloud enhancements is consistent with the tendency of the IFS model to overestimate high-cloud cover and ice water content (Bouniol et al., 2007).

The TB simulation stands out in that its cloud statistic greatly overestimates the occurrence and intensity of both above-cloud enhancements and below-cloud dimming. While the in-cloud frequency of occurrence for both dimming and enhancements provides a reasonable match with observation, the average intensity of these effects is much larger than observed.

## 4 Surface Observations

In this section, photolysis rate simulations are compared against surface measurements from the Chemistry-Aerosol Mediterranean Experiment 2013 (ChArMEx) in Lampedusa, CYprus PHotochemical EXperiment 2014 (CYPHEX) in Cyprus, and the Chilbolton site in England. Notable differences between these data sets are that the ChArMEx and CYPHEX campaigns took place during near clear-sky summertime conditions, whereas the Chilbolton measurements took place during cloudy wintertime conditions. A number of diagnostic model simulations are also included in this section, as listed in Table 3.



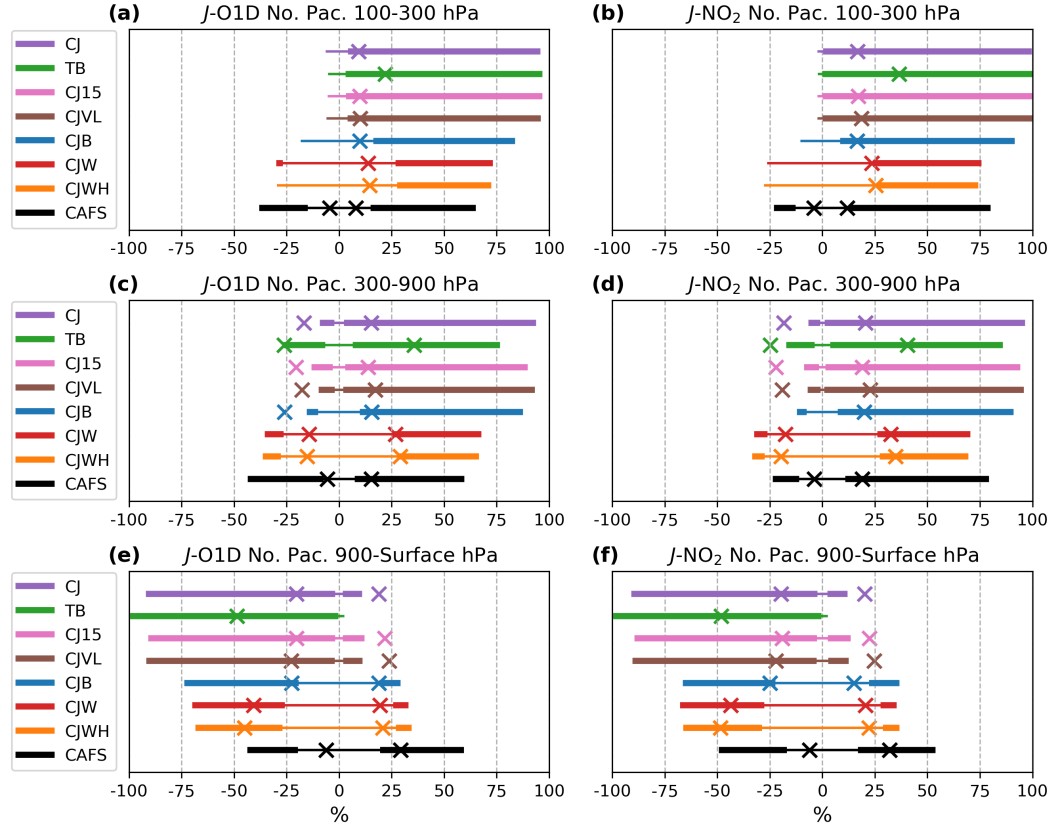

**Figure 4.** Frequency of occurrence and magnitude of the $J$-O($^1$D) (a-c) and $J$-NO$_2$ (d-f) change due to clouds for the below-cloud, in-cloud, and above-cloud tropospheric regions over the Northern Pacific basin. The interpretation of the line segments and marker locations is explained in the text, as are the legend labels.

**Table 3.** Diagnostic model configurations for the comparisons to surface observations.

| Name | Description |
|---|---|
| EMEP-A$_0$ | As EMEP-CJ (Section 2.3) but with zero surface albedo in the photolysis rate calculations |
| EMEP-A$_0$O$_3$ | As EMEP-A$_0$ but with no contribution of below-100 hPa O$_3$ to the photolysis rate calculations |
| EMEP-CSA$_0$ | As EMEP-A$_0$ but using clear-sky photolysis rates |
| EMEP-A$_0$HR | As EMEP-A$_0$ but using a high-resolution 0.1° x 0.1° horizontal grid |
| EMEP-CJBh | As EMEP-CJ but using hourly photolysis rate updates together with the Briegleb averaging cloud scheme (Section 5.3) |





## 4.1 ChArMEx 2013

During the ChArMEx 2013 campaign, surface $J$-O($^1$D) and $J$-NO$_2$ measurements were made at the ENEA Station for Climate Observations on the island of Lampedusa (35.5°N,12.6°E). The Lampedusa site, situated on a cliff at about 45 meters above sea level on the northeastern tip of the island, is considered a background site with relatively few sources of anthropogenic air pollution (Becagli et al., 2017; Calzolai et al., 2015; Casasanta et al., 2011). The measurements took place between DOY 157 (6th of June) and DOY 186 (6th of July) of 2013, with a detailed description of the campaign given by Mallet et al.

(2016). Atmospheric conditions within the Mediterranean basin and adjacent regions were characterised by lower than average temperatures, and limited biogenic and biomass burning aerosol emissions during the campaign. The moderate aerosol mass concentrations observed at the Lampedusa site were primarily controlled by marine aerosols, with PM$_{10}$ having an average value of 21 $\mu$g m$^{-3}$ over the course of the campaign (Mallet et al., 2016). However, diagnostic EMEP-CJ simulations find the impact of aerosols on the simulated photolysis rates to be small, showing an average noon-time dimming of 1.5%.

A detailed overview of the instrumentation used to measure $J$-O($^1$D) and $J$-NO$_2$ is given by Casasanta et al. (2011). In brief, downwelling actinic flux measurements from a METCON diode array spectrometer (DAS, Edwards and Monks, 2003) were used to measure the photolysis rates at 0.5-5 second integration times for local times between sunrise and sunset. A 2-$\pi$ input optics device was used to send incident photons to the spectroradiometer. The optics device was equipped with a matt black shadow ring, while also applying a stray light correction. The 1-$\sigma$ uncertainty was less than 10% for actinic fluxes below 300

nm, and less than 3-4% in the 300-340 nm range. The cross-section and quantum-yield data used to derive the $J$-O($^1$D) and $J$-NO$_2$ values from the actinic flux measurements are described in Casasanta et al. (2011) and Mailler et al. (2016), respectively. For the measurement-to-model comparison, the observations are averaged in hourly bins matching the averaging periods of the model output. Model output is sampled at the grid-box most closely representing the location of the measurement site. Further, since only photolysis by the downwelling actinic flux is measured, Cloud-$J$ photolysis rate calculations where the surface

albedo is set to zero (EMEP-A$_0$) are also included. It is worth noting that this does not consider the portion of the upwelling radiation redirected into the downwelling field from atmospheric scattering, although this contribution is likely of secondary importance. We also note that the surface albedo is set to zero only for the photolysis rate model output, whereas the photolysis rates used in the chemistry remain unchanged.

Fig. 5 shows the comparison against observation for the EMEP-CJ, EMEP-TB and EMEP-A$_0$ simulations between DOY

160 and 167. While this represents only a subset of the total measurement data, the results discussed in the following are largely independent of the choice of dates, owing to the generally stable meteorological conditions during the campaign. Nevertheless, a comparison of EMEP-A$_0$ against the full observational dataset is given in Fig. S2 of the supplementary material.

Fig. 5a illustrates that the observed $J$-O($^1$D) values shows considerable variations in their daily maxima. For example, the observed daily maximum is $2.54\times10^{-5}$ s$^{-1}$ and $2.98\times10^{-5}$ s$^{-1}$ on DOY 164 and 166, respectively, amounting to a 17.3%

variation. These variations are reproduced by EMEP-A$_0$, having values of $2.83\times10^{-5}$ s$^{-1}$ and $3.39\times10^{-5}$ on DOY 164 and 166, respectively, amounting to a 19.8% variation. For all the hours shown in Fig. 5a, EMEP-A$_0$ over-predicts observation by 6.9% on average. In the EMEP-CJ simulation, the upward actinic flux from the surface reflection increases surface $J$-O($^1$D)



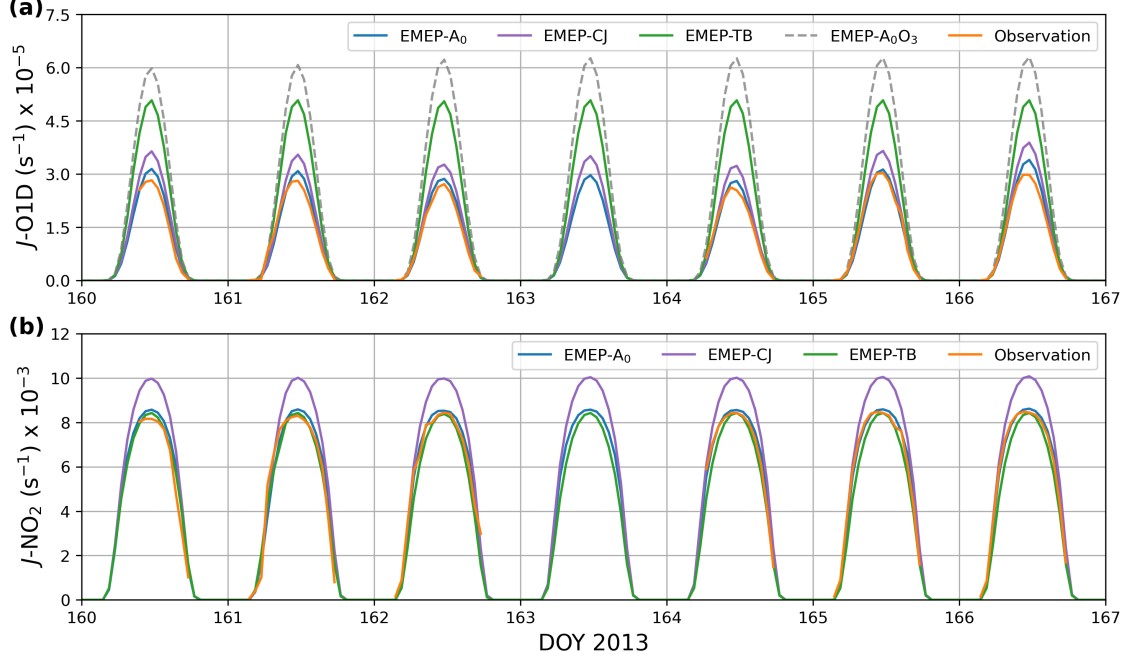

**Figure 5.** Hourly mean $J$-O1D **(a)** and $J$-NO$_2$ **(b)** photolysis rates observed at the Lampedusa site and simulated by EMEP-CJ and EMEP-TB model configurations between DOY 160 and 167 for the year 2013. The EMEP-A$_0$ simulation includes only the contribution from the downwelling surface actinic flux. In panel **(a)**, EMEP-A$_0$O$_3$ represents an EMEP-A$_0$ simulation where the contribution of the below-100 hPa O$_3$ column is excluded in the photolysis rate calculations.

rates by 40.9% on average, and by 14.8% for the noon-time values. The comparison between EMEP-TB and EMEP-CJ shows that the tabulated $J$-O($^1$D) values are considerably larger, with noon-time values being greater by 44.2% on average. We note that, since the surface albedo effect cannot be excluded in the tabulated photolysis rate scheme, the EMEP-TB results are most directly comparable to EMEP-CJ.

Fig. 5b compares the simulated and observed $J$-NO$_2$ values. Here both the EMEP-A$_0$ and EMEP-TB simulations closely follow the observed daily variations, which themselves show very little day to day variability. The EMEP-A$_0$ and EMEP-TB are both within 8% of the observed values on average. However, since the EMEP-TB rates include both the upwelling and downwelling actinic flux components, photolysis from the downwelling flux is likely to be underestimated. The EMEP-CJ simulation produces, as expected, values larger than observations, which do not contain the upwelling component.

### 4.1.1 TOC impact

In Fig. 5a, EMEP-A$_0$ Cloud-$J$ photolysis rate calculations where the contribution of the O$_3$ column below the EMEP 100 hPa model top is set to zero (EMEP-A$_0$O$_3$) are also included. The smaller day-to-day variability in this simulation indicates that the daily maximum $J$-O($^1$D) variations in EMEP-A$_0$ are largely caused by daily variations in the below-100 hPa O$_3$ column.

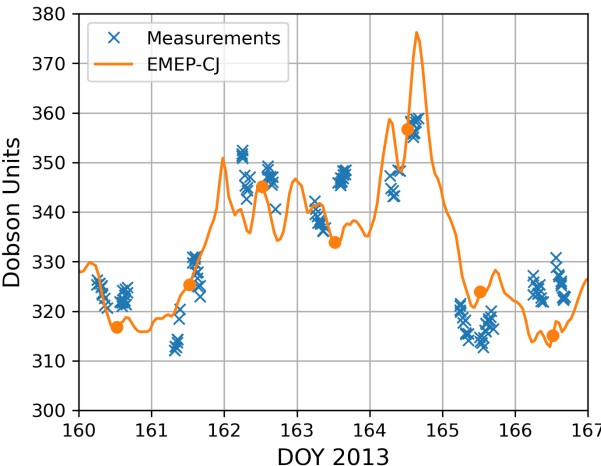

**Figure 6.** Total $O_3$ column (TOC) measurements (blue, cross) from the ChArMEx 2013 campaign. The orange line shows hourly total $O_3$ column values as simulated by the EMEP model appended with MEGRIDOP $O_3$ data above 100 hPa, with orange circles marking the noon-time values.

This is consistent with the variations in the observed and simulated TOC, as shown in Fig. 6. In this figure, TOC measurements made at the Lampedusa site by a Brewer spectrophotometer (Mailler et al., 2016; Meloni et al., 2005) during the ChArMEx campaign are compared against hourly modeled TOC. The orange dots highlight the noon-time simulated TOC, showing that this value was highest on DOY 164, coincident with the highest observed TOC and lowest observed and simulated $J$-O1D values. Similarly, the observed and simulated TOC are at a low on DOY 166, coincident with high observed and simulated $J$-O1D values. The contribution of the MEGRIDROP satellite $O_3$ measurements during the time period shown in Fig. 6 amounts to 247 DU. The $O_3$ column below 100 hPa thereby contributes between 20-30% of the TOC in the time period shown here, with the variations themselves likely caused by the variable contribution of lower stratospheric $O_3$. The much higher $J$-$O(^1D)$ model values obtained in EMEP-$A_0O_3$ further highlight the role of total $O_3$ in modulating surface UV radiation around 310 nm.

## 4.2 CYPHEX 2014

The CYPHEX 2014 campaign took place between July and August 2014 in Ineia, Cyprus, at a measurement site located about 600 meters above sea level (35.0°N,32.4°E). A detailed description of the CYPHEX campaign and measurement site is given by Mallik et al. (2018), Derstroff et al. (2017), and Meusel et al. (2016). In short, downward actinic flux measurements were made using a single monochromator spectral radiometer and 512 pixel CCD array as a detector (275–640 nm), having a total uncertainty of around 10%. The detector unit was attached to a 2-$\pi$ integrating hemispheric quartz dome, and photolysis rates for 8 of the reactions listed in Table 1 were calculated using the molecular parameters recommended by the International Union of Pure and Applied Chemistry (IUPAC) and NASA evaluation panels (Sander et al., 2011; IUPAC, 2015). In the current





work, 10-minute measurement data were averaged into hourly bins matching the EMEP model output. Given the prevailing
385    clear-sky conditions and low aerosol concentrations during the campaign (Derstroff et al., 2017), focus is placed on a few days
of measurements. Nevertheless, a comparison of EMEP-$A_0$ against the full observational data set is given in Fig. S3 of the
supplementary material.

Fig. 7 shows the comparison between observation and the EMEP-CJ, EMEP-$A_0$, and EMEP-TB simulations between DOY
186 and 191 (July 8th to 12th), for the eight measured photolysis reaction rates present in EmChem19. Focusing on $J$-O($^1$D)
shown in Fig. 7a first, EMEP-$A_0$ shows the closest agreement with observation for DOY 186 and 187. For the other days,
however, EMEP-CJ is closer to observation. The EMEP-TB values are higher than those simulated by EMEP-CJ, and show
moderate day-to-day variability attributable to light cloud cover variations in the input meteorology. The impact of the latter is
much smaller in the Cloud-$J$ simulations, however, with the calculated photolysis rates showing little day-to-day variability.
The other Cloud-$J$ photolysis rates show similarly little day-to-day variation, with the exception of $J$-NO$_3$. For the latter,
the impact of clouds is comparatively the largest both for model and observation, possibly owing to its almost exclusive
dependence on wavelengths in the visible spectrum. While EMEP-$A_0$ provides the general best agreement with observation,
EMEP-CJ consistently shows the closest agreement for $J$-H$_2$O$_2$ (Fig. 7f). On average, EMEP-$A_0$ underestimates $J$-H$_2$O$_2$ by
15.0% for the time period under consideration.

### 4.3 Chilbolton

Photolysis rate measurements were made between the 17th of December 2020 and 22nd of March 2023 at Chilbolton Observa-
tory, situated at a rural background air quality monitoring station ∼100 km south-west of London (51.1°N,1.3°W). A detailed
description of the Chilbolton site and experimental setup is given in Walker et al. (2022). In short, spectral actinic flux was
measured between 280–750 nm with a ∼1 nm resolution, using a CCD spectroradiometer coupled with a fibre-optic cable to
a single 2-$\pi$ quartz receiver optic. The station is located 78 meters above sea level, with the measurement device mounted 6.5
meters above ground level on a black-painted cabin roof railing. In addition to a dark signal correction, a stray light correction
was applied to account for any actinic flux originating from the surrounding grass and arable farmland. IUPAC-recommended
values for absorption cross-sections and quantum yields were used to derive photolysis rates at a 1-minute resolution, which
were used to construct a time series of hourly values in the current work. The agreement between CCD spectroradiometer
derived photolysis rates and those measured by a double-monochromator reference instrument was found to be within ±5%
(Bohn and Lohse, 2017). As before, only those measured photolysis rates that are present in EmChem19a are discussed.

During the measurement period, the 9th of March was the day most representative of clear-sky conditions (Walker et al.,
2022), with SZAs in the range of 50-60°. Fig. 8 shows the comparison against EMEP-$A_0$, EMEP-CJ, and EMEP-TB for this
day, while also including a comparison against EMEP-$A_0$ with clear-sky photolysis rates (EMEP-CSA$_0$). This figure shows
that EMEP-$A_0$ underestimates $J$-O($^1$D), $J$-HNO$_3$, $J$-H$_2$O$_2$, and $J$-CH$_3$CHO by a considerable margin, with the difference
being as large as a factor of two. The commonality between these photolysis rates is the large influence of absorbance at
wavelengths below 350 nm (Walker et al., 2022, Fig. 3), thus suggesting that the photolysis by UV radiation is underestimated.
It is unclear whether this model underestimation is the result of, for example, an overestimation of the total ozone column,



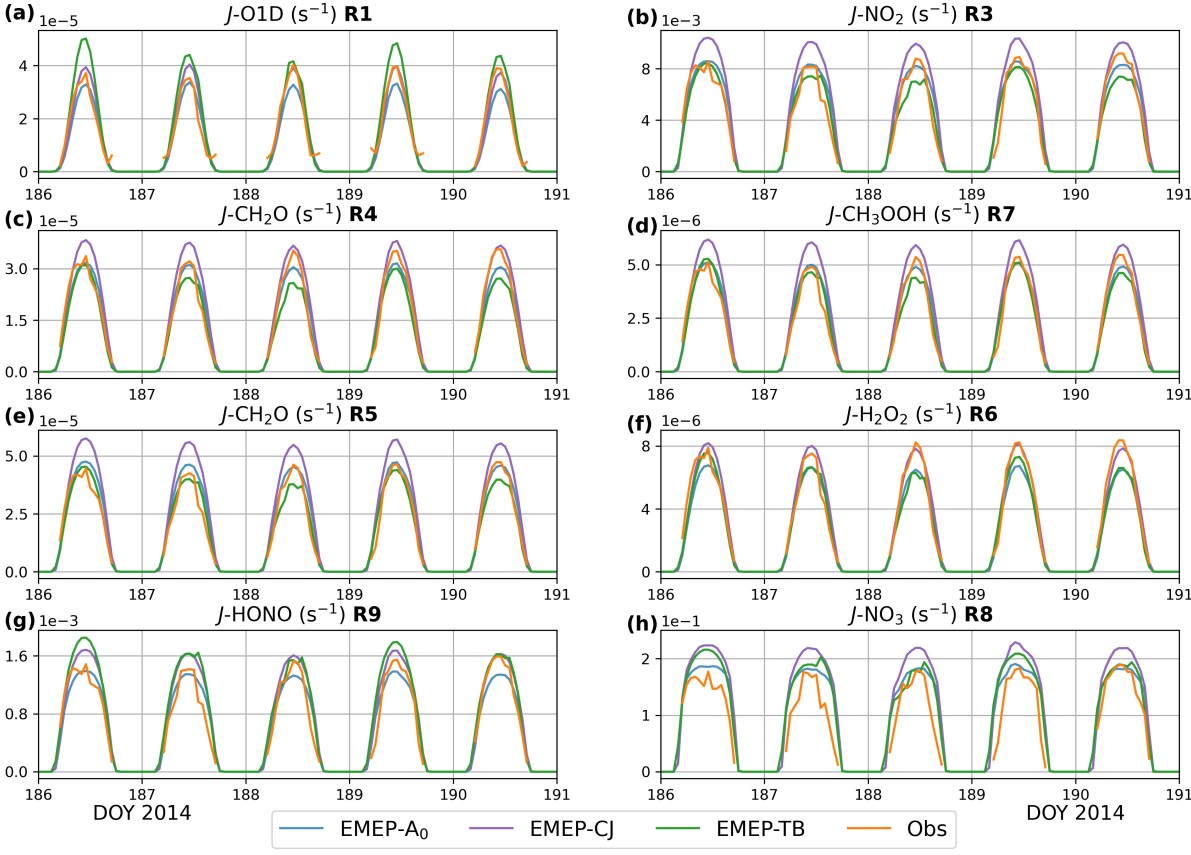

**Figure 7.** Hourly mean photolysis rates for eight photolysis reactions measured during the CYPHEX 2014 campaign (orange), and as simulated by EMEP-$A_0$ (blue), EMEP-CJ (purple), and EMEP-TB (green). Plot titles refer to the reactions listed in Table 1.

or of overestimated scattering or absorbance in the overhead atmospheric column. The close agreement between EMEP-$A_0$ and EMEP-CSA$_0$ shows that cloud cover and aerosols have very little impact on the Cloud-$J$ simulation results, except for the hours between 06:00-09:00 local time. However, the EMEP-TB photolysis rates decrease considerably after 12:00 hr local time, suggesting that the cloud effect is overestimated in the tabulated scheme.

Fig. 8 also shows that EMEP-$A_0$ overpredicts $J$-CH$_3$COCHO (MGLY) by a factor of around 2. This overprediction is similar to the results obtained using the Tropospheric Ultraviolet and Visible radiation model version 5.3 (TUV v5.3) by Walker et al. (2022), finding an overprediction of MGLY by a factor of $\sim$1.5. Since both the EMEP and TUV models employ the same cross-sectional and quantum-yield data as the Chilbolton spectral radiometer, based on Meller et al. (1991) and Chen et al. (2000), respectively, the results for MGLY hint at a possible model shortcoming elsewhere. One such shortcoming might be related to the large absorption cross-section component of MGLY in the visible light spectrum, where the Cloud-$J$ bin resolution is comparatively coarse.





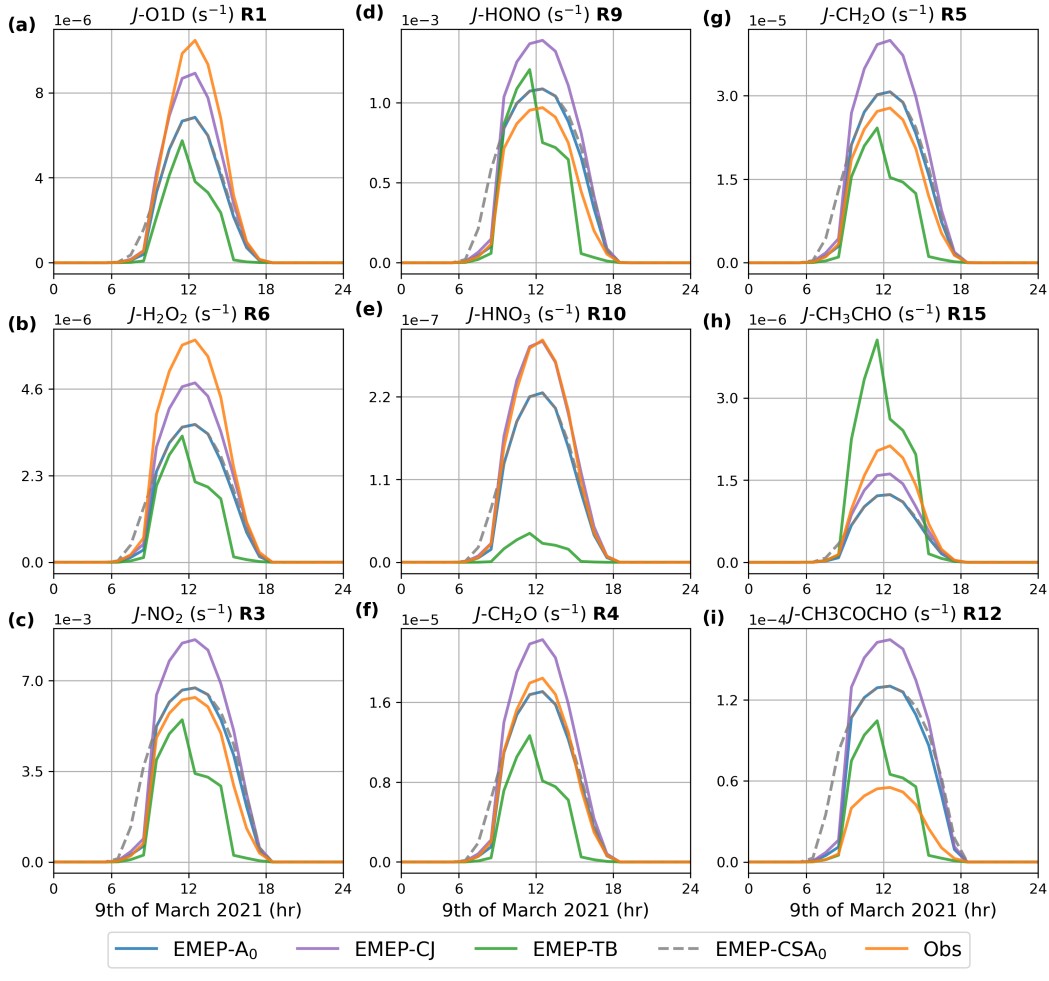

**Figure 8.** Hourly mean photolysis rates measured at the Chilbolton site (orange) and as simulated by the EMEP-$A_0$ (blue), EMEP-TB (green), EMEP-CJ (purple), and EMEP-CSA$_0$ (grey, dashed) model configurations for the 9th of March 2021. Plot titles refer to the reactions listed in Table 1.

As shown in Fig. 2 from Walker et al. (2022) and Fig. S4 of the supplementary material, the full time series of observed

and simulated photolysis rates display a large degree of day-to-day variability, superimposed on a seasonal variation. The observed day-to-day variations in peak photolysis rate values can largely be attributed to daily variations in local cloud cover and other cloud optical properties (Walker et al., 2022). In the following, the day-to-day variability is leveraged to investigate the representation of the simulated cloud effect. Fig. 9 shows scatter-plots of the observed and simulated EMEP-$A_0$ noon-time photolysis rate values, along with their correlation coefficients (r) and linear regression curves, for the 95 days of available

data. For reference, the correlation coefficients for the EMEP-CJ and EMEP-TB model configurations are also included. The day-to-day variability in peak photolysis rates is described well in EMEP-$A_0$, with correlation coefficients in the range of



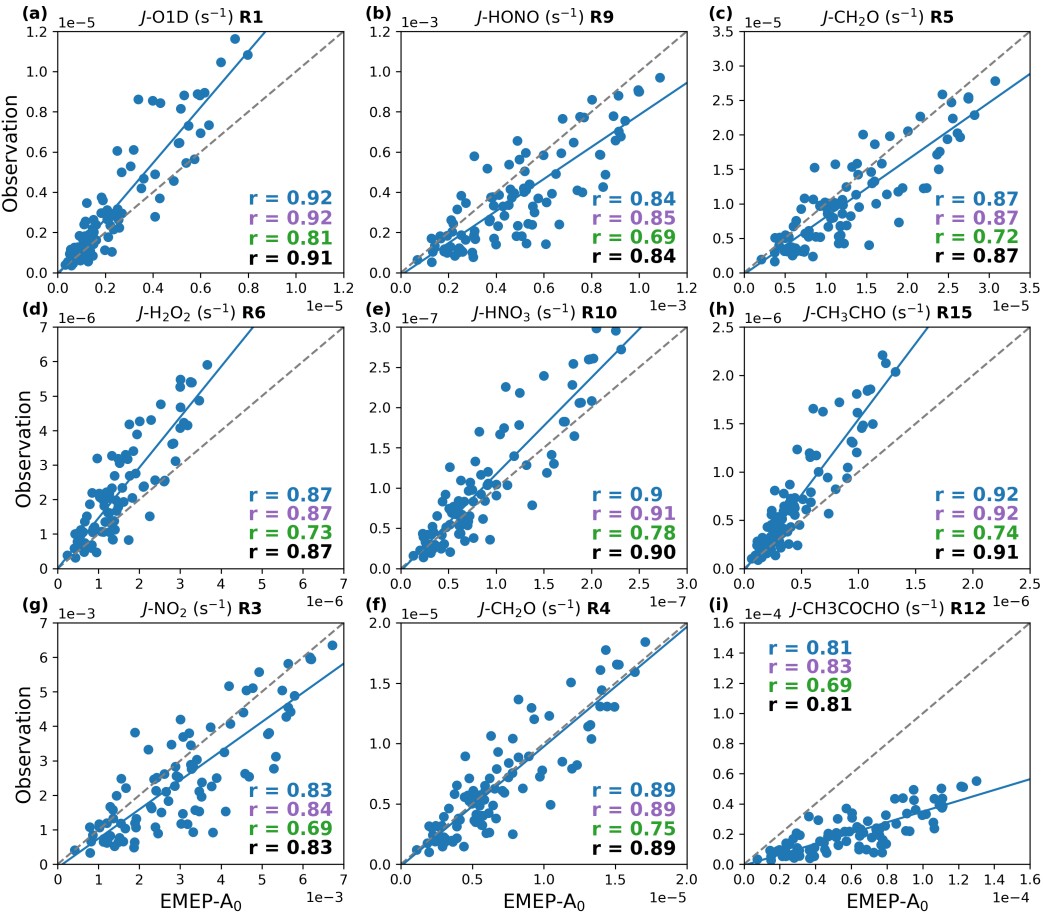

**Figure 9.** Noon-time hourly mean photolysis rates measured at the Chilbolton site and as simulated by EMEP-$A_0$ (blue) between the 17th of December and 22nd of March, 2021. The correlation coefficient (r) for EMEP-$A_0$, EMEP-CJ (purple), EMEP-TB (green), and EMEP-$A_0$HR (black) are also shown. Plot titles refer to the reactions listed in Table 1.

0.81-0.92. Similar results hold for the EMEP-CJ simulation, while the range of correlation coefficients is considerably lower for EMEP-TB (0.69-0.81). For reference, Fig. S4 also shows the normalized mean bias (NMB)(%) with respect to observation for the participating models, illustrating that the average values are considerably lower in EMEP-TB than in EMEP-CJ, with

the exception for $J$-CH$_3$CHO.

Fig. 9 and Fig. S4 also show the correlation coefficient and NMB statistics, respectively, for the diagnostic EMEP-$A_0$HR simulation. In this configuration, the EMEP-$A_0$ simulation is repeated on a higher regional 0.1° x 0.1° horizontal grid, leaving the other model parameters unchanged. Given the similarity between the EMEP-$A_0$ and EMEP-$A_0$HR results, the factor of 25 decrease in grid-box size is found to have little impact on the photolysis rate calculations. The values of the correlation

coefficients differ by no more than 0.01, while the NMB values show differences only on the order of a few percent points.





Similar diagnostic analysis finds that the horizontal grid resolution has a similarly small impact on the ChArMEx and CYPHEX measurement-to-model comparisons.





## 5 Model Analysis

This section investigates the impact of the Cloud-$J$ photolysis rates on the EMEP simulations results, using global simulations
for the year 2018. To this end, the oxidizing capacity of the troposphere is examined, in addition to the impact of the aerosol
direct effect. Model results for EMEP-CJ and EMEP-TB are also compared against surface observations of $O_3$, CO, and $NO_2$
across Europe, and against surface $O_3$ observations from four select hemispheric sites.

### 5.1 Oxidizing Capacity of the Troposphere

The tropospheric OH budget has historically been an important metric for the inter-comparison of global CTMs (Eastham
et al., 2014). Most natural and anthropogenic atmospheric gases are removed through oxidation initiated by OH, with the OH
abundance regulating the lifetime of key atmospheric species such as $CH_4$, VOCs, and CO (Lelieveld et al., 2016; Crutzen and
Zimmermann, 1991). Photolysis is the main driving mechanism for the production of OH, as the primary production pathway
of OH depends on the photolysis of $O_3$ (R2, Table 1) and the subsequent reaction of $O(^1D)$ with water vapour. Once produced,
approximately 75% of OH is lost to the reaction with CO (Crutzen and Zimmermann, 1991).

Following the approach recommended by Lawrence et al. (2001), Fig. 10 shows the air mass weighted annual mean zonal
mean OH budget across 12 tropospheric latitude-altitude sectors. Here, the results for EMEP-CJ and EMEP-TB are shown,
along with the results for the 17 global models participating in the inter-comparison study of Naik et al. (2013). In addition,
the reference climatology from Spivakovsky et al. (2000) is included. While the EMEP results are shown for the year 2018,
inter-annual variability of the yearly mean global OH abundance is on the order of 1-3% (Montzka et al., 2011), making the
results representative also for other years of simulation.

Fig. 10 shows that both EMEP-CJ and EMEP-TB simulate comparatively low OH concentrations in the mid- and high-
latitude upper troposphere (>500hPa), having OH concentrations near to or less than one standard deviation from the 17-
model mean. In the tropical upper troposphere, both EMEP-CJ and EMEP-TB fall within one multi-model standard deviation,
while EMEP-CJ shows the closest correspondence with the reference climatology. For the middle troposphere (750-500 hPa),
both EMEP-CJ and EMEP-TB predict concentrations near to the reference climatology, whereas the 17-model mean shows
considerably lower concentrations in the tropics.

The Cloud-$J$ photolysis rates considerably increases OH relative to EMEP-TB in the tropical lower troposphere (<750
hPa), with mass-weighted concentrations substantially exceeding the 17-model mean and reference climatology. In contrast,
the tropical lower atmosphere results for EMEP-TB are on the lower end of the 17-model results, while being close to the
reference climatology. This seemingly superior performance of the EMEP-TB model is, however, likely to be the result of
its underestimated tropical $J$-$O(^1D)$ values, as discussed in Section 3. EMEP-CJ nevertheless brings the global average mass-
weighted OH concentrations (11.67 x $10^5$ molec cm$^{-3}$) close to that of the multi-model mean reported by Naik et al. (2013)
(11.16 ±1.6 x $10^5$ molec cm$^{-3}$), compared to the 9.27 x $10^5$ molec cm$^{-3}$ average from EMEP-TB. For the overestimated
lower tropical OH in EMEP-CJ, possible sources may be related to, for example, excessive water vapour concentrations in the



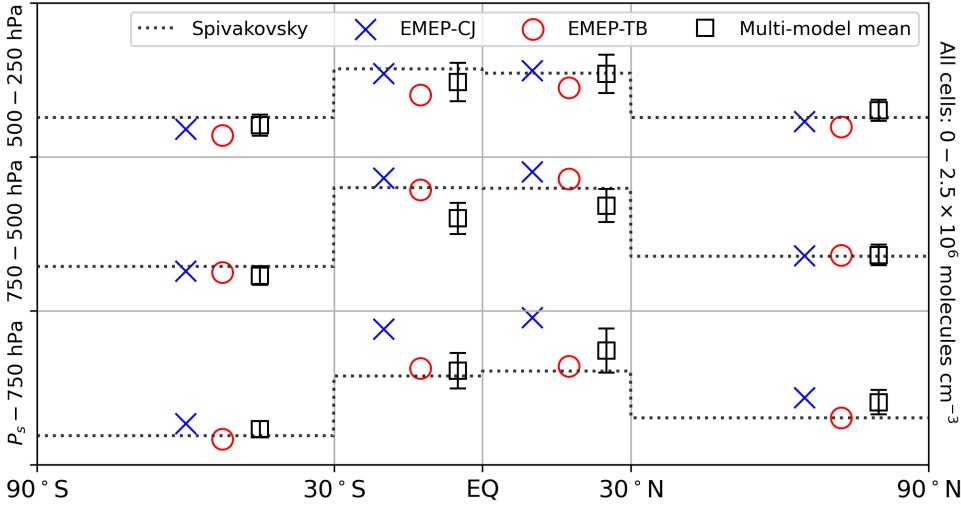

**Figure 10.** Mass weighted regional mean annual mean OH concentration from EMEP-CJ (cross), EMEP-TB (circle), the 17 multi-model mean and standard deviation from Naik et al. (2013) (square), and reference climatology from Spivakovsky et al. (2000) (dotted). The vertical range of each of the regions extends from 0 to 2.5 x $10^6$ molec cm-3.

input meteorology, excessive tropospheric ozone concentrations, or an underestimation of tropical CO concentrations (Travis et al., 2020). A detailed investigation is, however, beyond the scope of the current work.

### 5.2 Aerosol Direct Effect

Following the approach of Bian et al. (2003), changes to tropospheric $O_3$ concentrations are used as a marker of the impact of aerosol radiative scattering and absorption. To this end, monthly mean $O_3$ concentrations simulated by EMEP-CJ are compared

against those simulated by a diagnostic simulation where Cloud-*J* aerosol radiative effects are turned off (control run). Fig. 11 shows the difference between the two simulations at 250 and 2800 meters altitude for January and July 2018, calculated as EMEP-CJ minus the control run.

Fig. 11 shows that the $O_3$ perturbation caused by aerosols is largest over the tropical biomass burning regions. In July, biomass burning aerosol radiative effects reduce $O_3$ concentrations by as much as 12-16 ppb over central Africa. While this

impact is a factor of 2-3 stronger than the climatological findings of Bian et al. (2003), the EMEP-CJ region affected by biomass burning is also comparatively more geographically contained, possibly owing to the single year of simulation. In EMEP-CJ, both sea-salt and dust impact $O_3$ by less than 1-2 ppb, which for dust represents a smaller effect than the 3-4 ppb $O_3$ decrease over the Middle East and parts of Southern Europe reported by Bian et al. (2003) for July conditions. While the impact of dust in the EMEP-CJ configuration is smaller than that reported by Bian et al. (2003), the reason for this can be easily obscured

by its dependence on the simulated aerosol concentrations, aerosol chemical properties, calculated aerosol optical properties, and radiative transfer scheme (Real and Sartelet, 2011; Kinne et al., 2003). Assessing the importance of the aforementioned





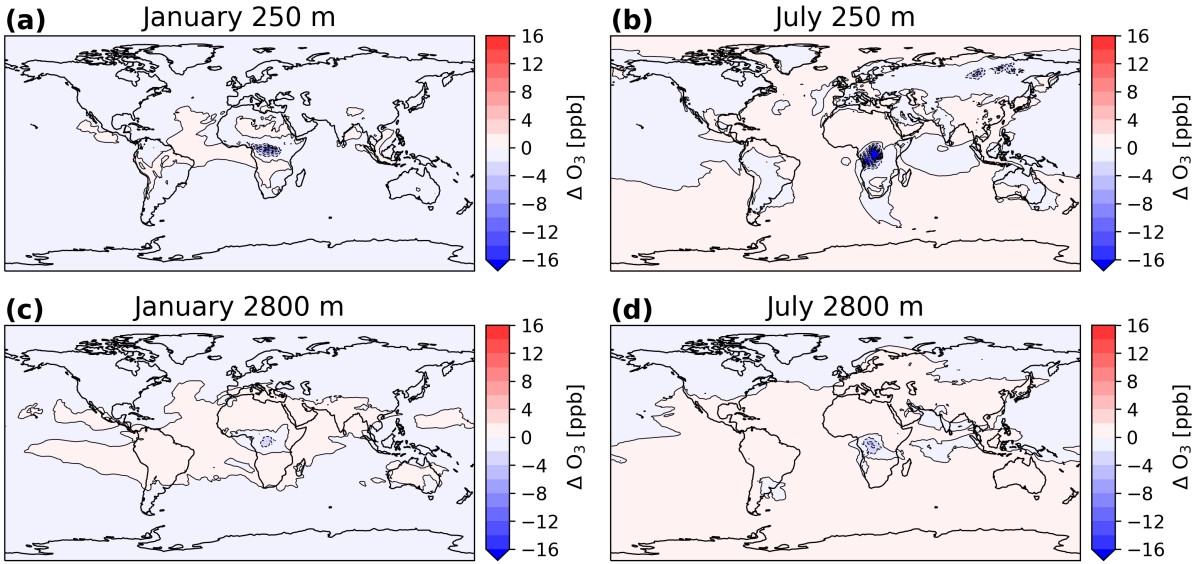

**Figure 11.** Perturbation to monthly mean O$_3$ concentrations at 250 and 2800 meters altitude in January **(a,b)** and July **(c,d)** 2018 due to aerosol radiative effects.

effects within the EMEP-CJ model would therefore require a coordinated approach combining other models with a detailed comparison against observed aerosol properties.

### 5.3 Surface Air Pollution

In this section, the EMEP-CJ and EMEP-TB models are evaluated against O$_3$, NO$_2$, and CO surface observations from the EBAS database (Tørseth et al., 2012). The data used comprises hourly time-series from 138 stations for O$_3$, and daily mean time-series from 93 and 8 stations for NO$_2$ and CO, respectively. The measurement stations are spread across urban, suburban, and rural sites throughout Europe. In addition, the hourly O$_3$ time-series are used to construct a time-series of daily O$_3$max concentrations, being the maximum 1-hourly mean value over the course of a day, which is a commonly used indicator for air

quality. Table. 4 shows the normalized mean bias (NMB)(%) and correlation coefficient between the models and observations for the winter (DJF), spring (MAM), summer (JJA), and fall (SON) seasons, as well as for the annual mean. Here all statistics are averaged over their respective time periods and across all stations, using the measured and modeled daily values. The time-series of the data underlying statistics shown in Table. 4 is included in Fig. S5 of the supplementary material.

Table. 4 highlights that O$_3$ and O$_3$max concentrations are generally increased in EMEP-CJ, while NO$_2$ and CO concentrations

are decreased. The use of EMEP-CJ thereby has the clear consequence of partitioning more Ox (=O$_3$ + NO$_2$) into the O$_3$ component. The correlation coefficients are most strongly impacted during the spring and summer seasons, with the correlation coefficients for CO and O$_3$ seeing the largest increases of 0.03 in spring and summer, respectively. The positive bias in the daily mean O$_3$, together with the generally low bias for O$_3$max, indicate that the diurnal variations are underestimated in



**Table 4.** NMB (%) and correlation coefficients for EMEP-CJ and EMEP-TB (brackets) against surface observations from the EBAS database in Europe. Statistics are based on daily values, and are shown for the four seasons as well as for the yearly average.

| Species [ppb] | Statistic | DJF | MAM | JJA | SON | Yearly |
|---|---|---|---|---|---|---|
| $O_3$max | NMB | 2.4 (-8.7) | 6.0 (-6.2) | 0.0 (-10.2) | 7.9 (-6.6) | 3.9 (-7.9) |
| | r | 0.58 (0.60) | 0.62 (0.62) | 0.79 (0.77) | 0.80 (0.79) | 0.80 (0.79) |
| $O_3$ | NMB | 5.3 (-7.7) | 10.4 (-3.0) | 6.6 (-4.4) | 15.0 (-0.9) | 9.2 (-3.9) |
| | r | 0.64 (0.65) | 0.54 (0.56) | 0.73 (0.70) | 0.72 (0.73) | 0.74 ( 0.74) |
| $NO_2$ | NMB | 3.5 (12.2) | -19.6 (-7.1) | -14.4 (-3.4) | -11.8 (-2.4) | -9.8 (0.4) |
| | r | 0.63 (0.64) | 0.68 (0.69) | 0.62 (0.63) | 0.72 (0.73) | 0.67 (0.68) |
| CO | NMB | -9.5 (-3.4) | -8.8 (3.4) | -9.4 (4.0) | -4.7 (3.3) | -8.1 (1.5) |
| | r | 0.64 (0.64) | 0.67 (0.64) | 0.72 (0.71) | 0.72 (0.72) | 0.70 (0.70) |

EMEP-CJ. For $NO_2$, the largest absolute NMB percentage changes occur in spring and summer, with EMEP-CJ considerably

underestimating $NO_2$ concentrations. Note however, that EMEP-TB also underestimates $NO_2$ in summer, and that EMEP-TB overestimates $NO_2$ in winter. The NMB for CO is generally increased, even though its correlation coefficient is increased in spring and summer.

Fig. 12 highlights the spatial variations in the yearly mean $O_3$max NMB changes between EMEP-CJ and EMEP-TB. In this figure, the difference between the absolute annual mean NMB at the measurement sites is shown, with decreases (increases)

in NMB marked by blue (red) dots. Overall, EMEP-CJ has annual mean $O_3$max values closer to observation across most of Central and Eastern Europe, while the performance is decreased along the Western coast of Spain, Portugal, and the United Kingdom. The latter is indicative of a possible overestimation of the influx of $O_3$ carried over the Atlantic Ocean by the surface Westerlies, noting that the NMB in these regions is generally greater than zero.

We note that the results presented in this section are largely independent of using either the MAX-COR AvQCA or Briegleb

averaging cloud schemes (as discussed in Section 2.1), in addition to updating the photolysis rates only once every model hour rather than model time-step (EMEP-CJBh, Table 3). Employing model time-step calculations with the MAX-COR AvQCA scheme, the EMEP model run-time is increased by roughly 250% compared to EMEP-TB. Using an hourly update approach combined with Briegleb averaging, however, the run-time increase is only about 15%. The correlation coefficients and the NMB shown in Table 4 are nevertheless nearly identical for EMEP-CJBh relative to EMEP-CJ, as shown in Table S1 from the

supplementary material. The correlation coefficients are changed by no more than 0.01, and only for $O_3$max in summer, while the NMB changes by no more than 0.8%. For reference, the time-series of surface pollutants calculated using EMEP-CJBh is included in Fig. S5. Similar results for using either the MAX-COR AvQCA or hourly Briegleb approach also hold for the higher resolution 0.1° by 0.1° grid.

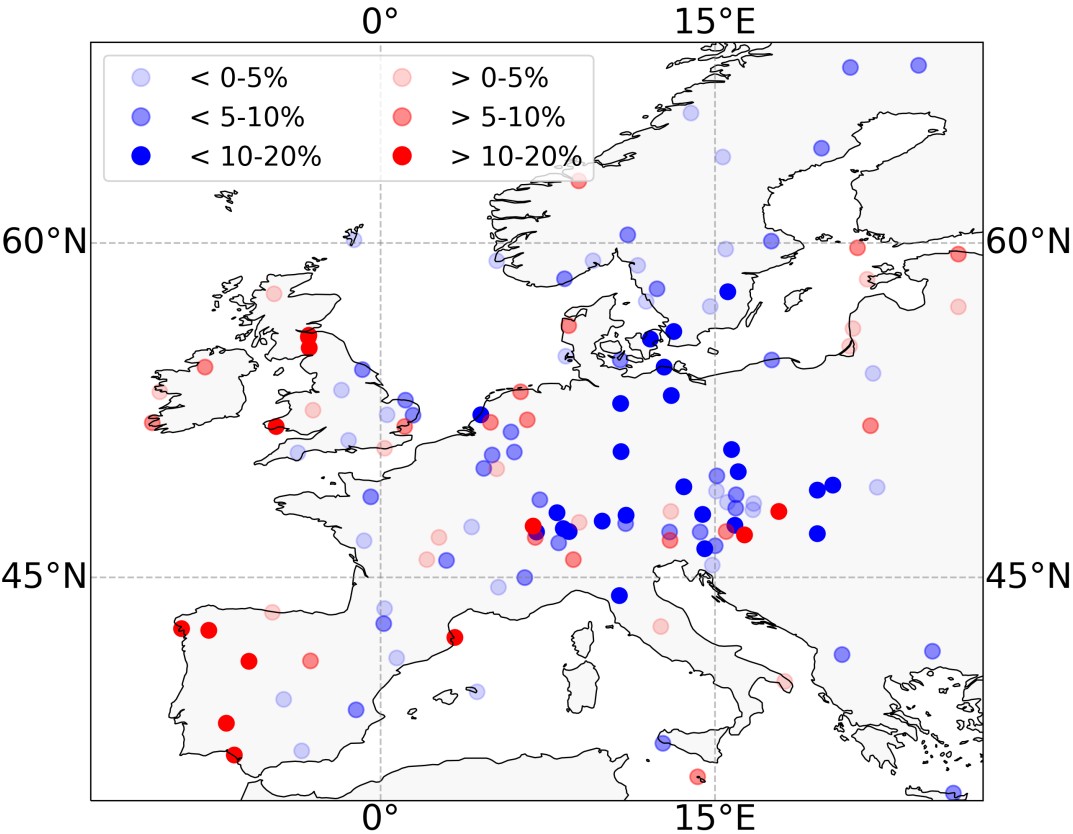

**Figure 12.** Difference in the absolute yearly mean O$_3$max NMB (%) between the EMEP-CJ and EMEP-TB simulations relative to surface observations from the EBAS database. Blue dots indicate that the average simulated value is closer to observation in EMEP-CJ than in EMEP-TB, and vice versa for the red dots.

### 5.3.1 Global Sites

This section compares simulated daily maximum O$_3$ concentrations against observations from four Global Atmospheric Watch (GAW; Schultz et al., 2015) stations, from locations representative of four distinct hemispheric air masses. To that end, the Ryori site (39.0°N, 141.8°E) in Japan is chosen as being indicative of O$_3$ production downwind of mainland China, and the Tudor Hill site (32.3°N, 64.9°W) is chosen as being indicative of O$_3$ production downwind from the United States. The seasonal variations at the Mace Head (53.3°N, 9.9°W) and Trinidad Head (41.1°N, 124.2°W) stations are taken to be broadly representative of background air masses carried over the North Atlantic and Pacific oceans, respectively (Parrish et al., 2009).

Fig. 13 shows the time-series of observed and modeled daily maximum O$_3$ concentrations. The observed seasonal and daily variations are described well in both the EMEP-CJ and EMEP-TB model configurations, although the daily variations during summertime are overestimated at Trinidad Head in both model configurations. Consistent with the results from the



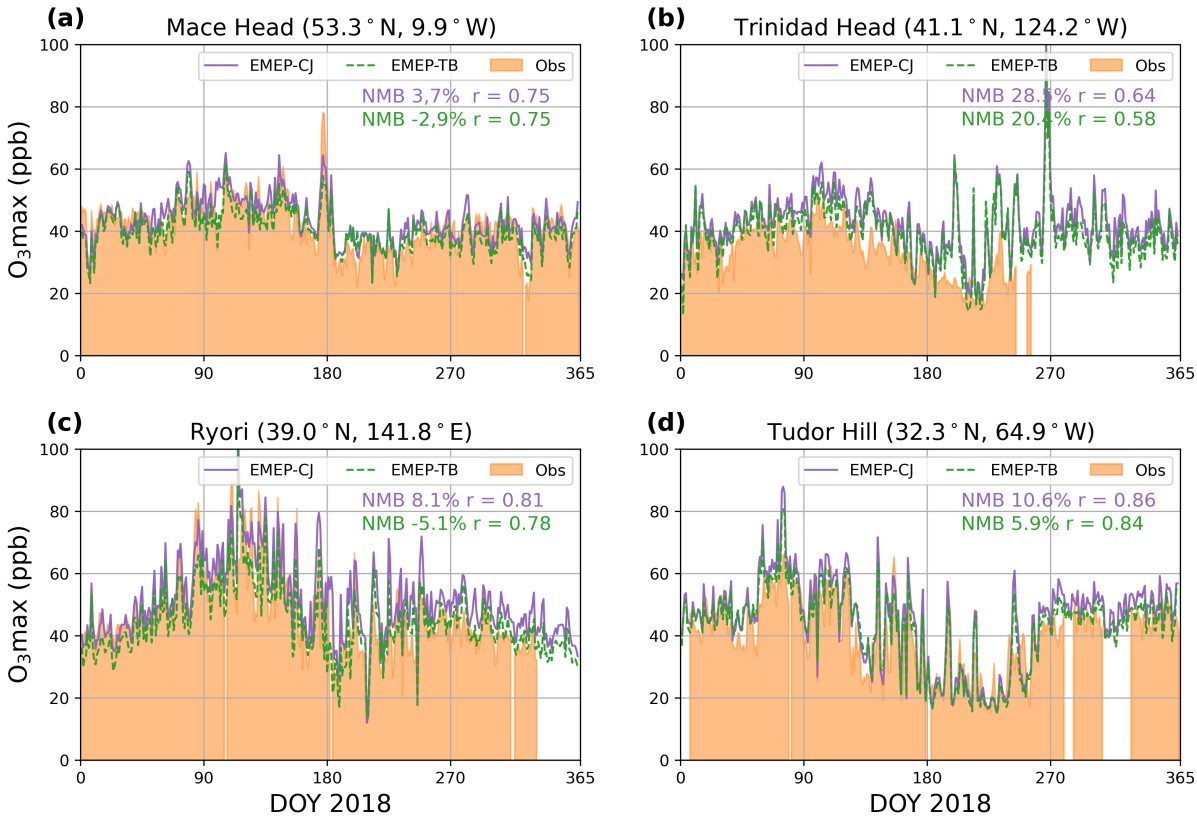

**Figure 13.** Time-series of observed and simulated daily maximum $O_3$ concentrations at four northern hemisphere GAW sites. The shaded area refers to the observed surface station concentrations, whereas the green dashed line refers to EMEP-TB and the purple solid lines to EMEP-CJ. The model NMB (%) and correlation coefficients with respect to observation are shown in each of the panels.

previous section, surface $O_3$max concentrations are generally higher in EMEP-CJ than in EMEP-TB. For the stations upwind
and downwind of the United States, this further increases the positive bias present in EMEP-TB. For the Mace Head and Ryori stations, the negative bias of EMEP-TB is replaced by a positive bias of a similar magnitude. EMEP-CJ increases the correlation coefficient by up to 0.06 compared to EMEP-TB, although it remains unchanged at the Mace Head station.



## 6 Conclusion

This study implemented and evaluated the Cloud-$J$ v7.3 photolysis rate calculation code in the EMEP model. Cloud-$J$ repre-
sent a major update with respect to the old tabulated photolysis rate system, offering an improved representation of the cloud
radiative effect, in addition to providing a description of aerosol radiative effects. The efficacy of Cloud-$J$ is established through
its favourable and robust comparison against measurements from the CAFS instrument onboard the ATom-1 flight campaign
over the Pacific Ocean, and against surface measurements from the Lampedusa (CharMex 2013), Cyprus (CYPHEX 2014),
and Chilbolton surface sites. The old tabulated scheme is found to greatly overestimate the cloud radiative effect, while also
underestimating clear-sky photolysis rates in the tropics. While the general performance of Cloud-$J$ is encouraging, the Cloud-
$J$ based simulations show a tendency to overpredict the frequency of occurrence and magnitude of above-cloud enhancements
and below-cloud diminishments, while also underestimating wintertime photolysis rates sensitive to wavelengths below 350
nm.

Notable results from the sensitivity analysis are that the calculated Cloud-$J$ photolysis rates are comparatively insensitive to
the temporal, vertical, and horizontal model resolution. They are instead more strongly impacted by the difference in the cloud
field representation between the IFS and WRF models, and to a lesser extent by the choice of cloud effect scheme between
the MAX-COR AvQCA and Briegleb averaging schemes. However, using the comparatively less computationally expensive
Briegleb averaging scheme, together with hourly photolysis rate updates, is found to have a negligibly small impact on the
EMEP simulation results, while keeping the total model run-time increase below 15%. In the comparison to the Lampedusa
surface observations, diagnostic simulations further find that daily variations in the total $O_3$ column induces up to 17.3%
(19.8%) variations in measured (modeled) day-to-day maximum surface $J$-$O(^1D)$ values.

The global model results with Cloud-$J$ enabled show a general improvement in the correlation and NMB statistics with
respect to surface daily maximum $O_3$ concentrations across Europe. The model bias for surface $NO_2$ is significantly worsened
during spring, however, possibly hinting at model shortcomings elsewhere, given the improvement in $J$-values. In addition,
a negative bias for surface CO is introduced, even though its correlation coefficient increases during the spring and summer
seasons. The comparison against four northern hemispheric background surface measurement sites, confirms the finding that
using Cloud-$J$ leads to generally higher simulated surface $O_3$ concentrations across the Northern hemisphere. On a global
scale, Cloud-$J$ increases the annual mass-weighted tropospheric OH budget from 9.27 x $10^5$ molec cm$^{-3}$ to 11.67 x $10^5$ molec
cm$^{-3}$, bringing it in line with the 11.12 x $10^5$ molec cm$^{-3}$ 17-model mean reported by Naik et al. (2013). The OH budget
of the tropical lower atmosphere is increased sharply, however, with the new values of OH likely to be overestimated. The
simulated aerosol direct effect is found to mostly reduce $O_3$ concentrations over large biomass burning regions in the tropics.

While the focus of the current work lies on the implementation of the Cloud-$J$ code within the EMEP model, the demon-
strated performance of Cloud-$J$ also builds confidence in its use in the wider range of models currently employing it or one
of its predecessor codes. The sensitivity analysis presented in this work also aids the interpretation of the large inter-model
spread in photolysis rates reported by Hall et al. (2018). For future work, the analysis presented in this work can be extended
to include the impact of Cloud-$J$ on the EMEP model performance during isolated events, such as dust storms and ozone





episodes. Future work can also include a description of the direct photolysis of Secondary Organic Aerosol (SOA) particles, which has been shown to cause reductions of up to 50% in biogenic SOA loading (Zawadowicz et al., 2020).





*Code and data availability.* The ECLIPSE global emission inventory is available from https://iiasa.ac.at/web/home/research/researchPrograms/
air/ECLIPSEv6b.html. Last access May 2023.

The MEGRIDOP dataset is available from https://cds.climate.copernicus.eu/cdsapp#!/dataset/10.24381/cds.4ebfe4eb?tab=overview. Last access May 2023.

The ATom-1 observation and simulation data are available from https://daac.ornl.gov/ATOM/guides/ATom_Photolysis_Rates.html, archived as part of Hall et al. (2019). Last access May 2023.

The EBAS data are available from https://ebas.nilu.no/. Last access May 2023.

The model and measurement data, model source code, and Python scripts used in this work are available from https://zenodo.org/record/8282997 (van caspel et al., 2023)

.



# Appendix A

## A1   CRI v2-R5

The Cloud-*J* code has also been implemented in the EMEP model version running with the CRIv2R5 chemical mechanism. The CRIv2R5 mechanism is based on the CRI v2 reduced (lumped chemistry) scheme of intermediate complexity, which is traceable to the Master Chemical Mechanism (MCM) v3.1 (Utembe et al., 2011; Jenkin et al., 2008). CRIv2R5 was developed by systematically testing and lumping anthropogenic VOC emissions (Watson et al., 2008), with non-methane VOCs being represented by 22 compounds. For almost all photolysis reactions, Cloud-*J* cross-sectional data are publicly available from the GEOS-Chem v14.1.1 model (GEOS-Chem, 2023). However, the GEOS-Chem data are extended for the *n*-butanal and *i*-butyraldehyde species present in CRIv2R5. For this, cross-sectional and quantum yield data from Martinez et al. (1992) and Tadić et al. (2001) are used for *n*-butanal, and cross-sectional and quantum yield data from Martinez et al. (1992) and Chen et al. (2002) are used for *i*-butyraldehyde, respectively.



*Author contributions.* WEC developed the model code, performed the analysis, and wrote the manuscript. DS and JEJ oversaw the Cloud-J implementation in the EMEP and BoxChem models. AMKB helped to prepare specific IFS input meteorology files, while MV prepared the WRF input files. YG helped with the photolysis rate update in the CRIv2R5 chemical mechanism. AS provided the ChArMEx photolysis measurements, while GP provided the corresponding column ozone measurements. HLW and MRH provided the Chilbolton photolysis measurements. All authors contributed to the discussion and review of the manuscript.

*Competing interests.* The authors declare that no competing interests are present.

*Acknowledgements.* This work has been in part funded by the EMEP Trust Fund. IT infrastructure in general was available through the Norwegian Meteorological Institute (MET Norway). Some computations were performed on resources provided by UNINETT Sigma2 - the National Infrastructure for High Performance Computing and Data Storage in Norway (grant NN2890k and NS9005k).

We acknowledge all researchers and supporting personnel who participated in the CYPHEX 2014 and ChArMEx 2013 campaigns. The
authors also acknowledge the support of the personnel involved with the measurements at the Chilbolton Observatory.

The EBAS database has largely been funded by the UN-ECE CLRTAP (EMEP), AMAP and through NILU internal resources. Specific developments have been possible due to projects like EUSAAR (EU-FP5)(EBAS web interface), EBAS-Online (Norwegian Research Council INFRA) (upgrading of database platform) and HTAP (European Commission DG-ENV)(import and export routines to build a secondary repository in support of www.htap.org). A large number of specific projects have supported development of data and meta data reporting
schemes in dialog with data providers (EU)(CREATE, ACTRIS and others). For a complete list of programmes and projects for which EBAS serves as a database, please consult the information box in the Framework filter of the web interface. These are all highly acknowledged for their support.

This work was in part supported by the UK Natural Environment Research Council award number NE/R016429/1 as part of the UK-SCAPE and in part by the NC-International programme [NE/X006247/1] delivering National Capability.
We would like to thank Michael J. Prather for making the Cloud-*J* model publicly available, and we are grateful for the code availability from Hall et al. (2019) for use in the ATom-1 analysis. We also acknowledge the use of Pyaerocom (Jonas et al., 2020, https://pyaerocom.met.no/) to compare the EBAS data against the simulations presented in this work.



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
