# Peer review of "Implementation and evaluation of updated photolysis rates in the EMEP MSC-W chemistry-transport model using Cloud-J v7.3e"

_Geoscientific Model Development, 2023_

## Author Response (AR1)

The authors would like to thank both reviewers for their thoughtful and detailed reviews, which have undoubtedly improved the quality of our work. Answers to both reviewers are organized into this single file.

In reply to the reviewers, some of the remarks have been combined into a single comment.

**Response to RC1 (Michael Prather)**

Comment 1:

L1 – 'chemical transport model' Sorry to be pedantic, but our community has had a long-standing problem with adjective-noun combinations. 'chemical' is an adjective and as such must be modifying 'transport', the following noun. Thus our model is a 'transport model' that transports chemicals, where in reality it is also a 'chemistry' model. Hence, the correct terminology for CTM should be 'chemistry-transport model'. I take part responsibility for this mistaken acronym since the term 'CTM' was first coined by the Theoretical Studies and Modeling Working Group of the 1986 Global Tropospheric Chemistry Report (UCAR OIES Report 3). We needed an acronym in parallel with GCM and named it Chemical Transport Model, which has been hard to live down – but it did catch on.

Reply 1:

Thanks for pointing this out. The terminology has been adjusted throughout the manuscript, including in the manuscript title.

Comment 2:

L9-11 – the 20% daily variability makes sense but the hydroxyl budget increase of 26% then seems large. Try to keep bias errors or shifts separate from variances.

L11: to say that the OH budget 'increased by 26%' is totally confusing. Did the P and L terms increase by 26%? Did the mean OH increase by 26%? 'budget' is a vague term here.

Reply 2:

To clarify that 'budget' here refers to the oxidation capacity of the troposphere, as defined in the text, "total tropospheric hydroxyl budget" is changed to "annual mean mass-weighted tropospheric hydroxyl concentration". By clarifying that the hydroxyl variations refer to annual means, we also hope to have cleared up the distinction between the 20% daily variability in ozone photolysis rates (here implicitly referring to J-O1D), and the changes to the oxidizing capacity of the atmosphere.

Comment 3:

L10 and later - 'aerosol radiative effect' This get very confusing with regard to the climate system for which these models are also used. 'radiative' would normally mean Watts per meter squared, but here you mean the change in photolysis fields. Maybe 'aerosol photolytic effect'. Later you heavily use 'aerosol direct effect'. This term is used heavily in the chemistry & climate community to describe climate forcing, and is contrasted with the aerosol indirect effect (clouds) and semi-direct effect (clouds disappear when the air is warmer). Please find another term, maybe something with 'photolytic' in it.

**Reply 3:**

We agree that our terminology is confusing. The 'aerosol radiative effect' has been changed throughout the text, using alternatives such as 'aerosol-photolysis interactions', 'aerosol photolytic effect', and 'photolytic impact of aerosols'.

**Comment 4:**

L14 - 'The bias is worsened...' Not sure what you mean here. Do not hint at results in the abstract, just give the main ones.

**Reply 4:**

This statement has now been removed from the abstract.

**Comment 5:**

L17: 'emissions, meteorology, and solar radiation. Photolysis .....' List the big 3 first, then go on to solar.

**Reply 5:**

Done.

**Comment 6:**

L22: 'reflections' => reflectivity

L22: scattering and absorption applies to air molecules as well as clouds and aerosols.

**Reply 6:**

Changed to reflectivity, and included air molecules in the scattering and absorption statement.

**Comment 7:**

L31: 'often referred to as' sounds stilted, maybe just (J-values) or i.e., J-values,

**Reply 7:**

Done, changed to (J-values).

**Comment 8:**

L37: 'eight different approaches for averaging over overlapping, broken cloud fields."

**Reply 8:**

Changed.

Comment 9:

L54: 'As noted above,' – delete, unnecessary

Reply 9:

Done.

Comment 10:

L63: '3-hourly IFS O3 concentrations are now specified at the top…' You need full O3 profiles to get the columns correct. Later you talk about monthly mean O3 columns. Is this used to get STE folds? Where did the IFS O3 come from? is it assimilated?

Reply 10:

The O3 model top boundary conditions are taken from the IFS ERA5 reanalysis product, which is now clarified in the text. These boundary conditions are indeed prescribed to capture the effects of STE. The text describing the above-100-hPa O3 used in the Cloud-J calculations has also been clarified, as described below.

Comment 11:

L83: drop 'However', it does not help the reading of the sentence

L84: Please use 'S' or 'N' on all latitude ranges: 30N-90N (with the degrees, OK)

Reply 11:

Done and done (the latter also in other places throughout the text).

Comment 12:

L84-85: If you mirror the latitudes, if there not a seasonal shift? Also, tropical atmospheres are very different from mid-latitude atmospheres, so using 30N-40N to do the tropics is not particularly good. Since this is a just a description of what the lookup tables DID, it should be left as is. Nevertheless, it may explain why there is such a large difference in the global budgets (e.g. OH) with Cloud-J.

Reply 12:

The tabulated system indeed fully mirror the latitudes, meaning that for example the photolysis rates for the wintertime NH atmosphere were also applied to the summertime SH.

Comment 13:

L98: This could be explained better. Use of wbin=12 provides equally accurate J-values in the troposphere as the full wbin=18 versions of Cloud-J (Prather, 2015) and saves 33%. The wavelengths important in the troposphere are: 200-220 nm (upper tropical troposphere only) and > 291 nm.

This has been clarified in the text.

Comment 14:

L105: Correct but could be simpler: Rayleigh scattering (gases = air) and absorption by O2 and O3 only, not NO2 or other gases that do not affect the radiative transfer solution.

Reply 14:

Rephrased to "These binned molecular data are used in the propagation of the top of the atmosphere solar irradiance spectrum through each ICA, taking into account Rayleigh scattering and absorption by $O_2$ and $O_3$, in addition to aerosol and cloud radiative interactions and surface reflections.

Comment 15:

L106: surface albedo – not directly relevant to this paper, but Cloud-J v7.6 has added an ocean surface albedo that is wavelength and angle dependent.

Reply 15:

Good to know! We should remember to include this in any future update (ocean surface albedo is normally fixed to a value of 0.08 in the EMEP model).

Comment 16:

L119: 'aerosol direct effect' – as noted above, find a new name for this.

Reply 16:

See reply to Comment 3.

Comment 17:

L140: 'photolysis of O3 for wavelengths below 320 nm is an important loss mechanism for tropospheric O3,' ONLY when followed by O(1D) + H2O reaction (<10% of O(1D)). Also the quantum yield for O1D is <1 everywhere. Please correct.

Reply 17:

Rephrased to reflect that it is a net loss of O3 only when followed by O(1D) + H2O, and for when the photolysis of O3 yields O1D.

Comment 18:

L143: 'overhead stratospheric O3' – This is only partially correct, please be more precise: i.e., the overhead O3 column above 100 hPa…You must calculate the lowermost stratospheric O3 column at mid latitudes (100-200 hPa) with EMEP.

'stratospheric' has been rephrased to 'above 100 hPa' ozone here as well as in the following text.

Comment 19:

L147: Now what really are you doing in EMEP CTM? You said you take the assimilated IFS O3 as upper boundary at model top = 100 hPa = ~16 km. Here you say you use a monthly climatology down to 10 km ?? 100 hPa is not the tropopause poleward of 30 degrees.

Reply 19:

The manuscript text was vague, in that it wasn't immediately clear that the 10-50 km range reflects the altitude range of MEGRIDOP measurement data availability. These measurement data are subsequently interpolated onto a pressure grid extending from 100 hPa (EMEP model top) up to 1 hPa (using the pressure data on altitude levels as provided by the MEGRIDOP dataset), and combined with the simulated EMEP O3 concentrations below 100 hPa.

The text of this section has been revised to provide a clearer description of how the MEGRIDOP O3 data are used.

Comment 20:

L154: do you mean: 'a multi-year monthly zonal mean climatology' Please be more precise.

Reply:

The text has been clarified to reflect that we meant a multi-year monthly mean climatology.

Comment 21:

L156: this level of variability due to monthly zonal means in column ozone is an interesting finding.

Reply 21:

During the writing of this work we have investigated this in some more detail, linking the ozone column changes to inter-annual variations in the Brewer-Dobson circulation (modulated by the QBO), which is being prepared for future publication.

Comment 22:

L158-164: Nice idea to control the control the number of degrees of freedom. Why not test the different CJ cloud parameterizations with box Chem?

Reply 22:

That would be a nice follow-up from the clear-sky boxChem calculations, though setting this up for the 1D model would require some thought on our part. Though, given the simplifications of the boxChem

model (e.g., no aerosols), maybe it would also make sense to design a dedicated EMEP model setup to tests the cloud parameterizations in more detail at a later stage.

Comment 23:

L195: Very nice sensitivity study and analysis, this is helpful in understanding what is driving the difference

L208: This is an interesting result. The tabulated Js must have very bad reference calculations for low sun angles. Added point – in Cloud-J v7.6 (beyond current version here), we have added refraction in addition to spherical atmospheres, which further adds to J-values at low sun and past sunset (in the stratosphere).

Reply:

Good to know!

Comment 24:

L209: Table 1. This table is great. It is clear and shows some interesting diagnostics. There are some obvious messages here about the difference between Tab-J and Cloud-J that indicate that Tab-J has some very old cross sections and quantum yields.

- J-O1D fell -25% with Cloud-J, that should to first order greatly reduce OH, since O(1D)+H2O is the primary source of OH and H2O is presumably the same. I suspect that the quantum yields are old values.
- >>>>However, the plot of J-O1d (Fig. 1) shows that TB is much smaller than CJ ?? which is correct?
- J-NO2 increased 25%, that is large, not sure what happened there
- J-HNO3 appears to increase 4x, probably not a bid deal since OH+HNO3 is major sink.
- J-HNO4 increases 13x, probably missing the 1 micron photolysis found by Wennberg.
- "P-dep from 177-999 hPa" is correct, but probably add a footnote that this range in P is tied to a tropical lapse rate, so it is both P and T interpolation.

Reply 24:

Relative to the boxChem results (45 degrees N), the most relevant comparison of J-O1D between TB and CJ in Fig. 1 is panel (c), showing the N. Pacific results. In this figure, the TB rates can be seen to overtake the CJ values close to the surface, which is where BoxChem is evaluated. Notwithstanding other possible differences, such as sun-angle dependence, other difference between boxChem and the latitudinal extent of the N. Pacific block, and surface albedo, Fig. 1 seems consistent with Table 1 in that the TB surface J-O1D rates at 45 degrees North are greater than those from CJ.

We have also added a footnote to Table 1 about the P-range being tied to a tropical lapse rate, and so making it both P and T interpolated.

L209 Table 2: '16$^{th}$ of August 2015' should be simply 16 August 2015.

Reply 25:

Changed.

Comment 26:

L210: It is nice to see the extension of ATom analysis (Hall 2018) to EMEP, along with more detail. Nicely done. The use of 2015 cloud fields in addition to the 2016 data is very useful.

L227ff: This analysis of J-O1D and J-NO2 is excellent and quite interesting, definitely a high-point of the paper. One case I found very thought provoking was the doubled-vertical-resolution case (CJLVL). For this case, did you redo the EMEP model levels to match the IFS? Can you specific the native IFS grid (e.g., we use IFS cy38r1 at T159L60 for our CTM)? It looks like the only big difference between CJ and CJLVL is for J-NO2 N.Pacific, is that true? That would be the biggest cloud-affected J.Also, you found out that WRF has very different cloud fields since they have different cloud physics than the ECMWF model. It further demonstrates that clouds are hard to assimilate and not just determined by u,v,q,T.

Reply 26:

For this we indeed made the EMEP model follow the IFS levels below 100 hPa (the EMEP grid follows that of the input meteorology – in this case IFS – except for the lowest layer which is kept at ~90m thickness). This information has been clarified in the text.

The model configuration section includes information about the IFS model version (cycle 40r1), with the referenced work providing the complete IFS model description.

Our conclusion is also that the only big difference for CJLVL is for J-NO2 in the Northern Pacific.

Comment 27:

Fig.1: It looks like CAFS (ATom) is using fundamentally different NO2 cross sections and quantum yields. These are tabulated separately in JPL. It is not an interp/extrapolation problem with T because it is the same over all altitudes.

L256: Did you check the T difference and if it could change the O1D ? You should do that before speculating here since it is under your control.

Reply 27:

It was an oversight to not realize that the J-O1D changes could also be caused by a difference in the above-100-hPa O3 column, and not just temperature. The statement on the temperature dependence has been removed.

**Comment 28:**

L259: I agree that this is what the Fig. 1 shows, but it is opposite direction and smaller than shown in Table 1.

**Reply 28:**

We hope to have addressed what was seemingly a mismatch in reply to Comment 24. The TB underestimations referred to here relate to the middle and upper tropospheric rates, which are underestimated in all regions and for both J-values.

**Comment 29:**

L265: It is good to see CJW & CJWH match each other so well in both clear and all sky. I cannot understand why CJW&CJWH separate for J-NO2 in Fig.1. Can you explain it?

**Reply 29:**

The clear-sky J-NO2 rates between CJW and CJWH indeed seem to separate slightly for the N. Pacific region, for which we have no explanation other than possible numerical noise, given the small differences in their J-values. The WRF model version used for generating the meteorological data sets was identical.

**Comment 30:**

L270 Fig.3: You want to use CJB because of the lower cost, so you may want to do a more thorough evaluation of the errors in CJB (blue) vs CJ (purple). Comparing with ATom here is very good, but you must also recognize that given the cloud distributions (and a non-3D RT solution), CJ is much closer to the correct answer.

**Reply 30:**

To add nuance to our results, we have added the following sentence to section 3.1:

"We note that, while the CJB results represent a comparatively large deviation from the more accurate CJ solution (given the IFS-based cloud distribution), the impact of using CJB on the EMEP simulation results will be discussed in more detail in Section 5.3."

While we acknowledge that this does not constitute a more thorough evaluation of the photolysis-rate errors, the statistical properties between CJB and CJ are discussed in more detail in the following section (section 3.2).

**Comment 31:**

→ CJB has consistently lower Js above 850 hPa and higher Js in the marine BL, esp. in the Tropics.

L275-279: Some caution here. ATom cannot fly in the MBL when there are thick clouds present. – there must always be clear enough patches to guarantee VFR when entering and leaving the MLB.

Unfortunately in Hall 2018, we could not /did not filter the model MBL cumulus layers to remove those conditions (did not know how to). Thus ATom=CAFS would have higher than average 950-hPa Js. You should probably revise this text.

**Reply 31:**

Thanks for sharing this insight. The text has been revised to now read:

"One caveat is that the ATom aircraft was unable to fly in the marine boundary layer under thick cloudy conditions due to flight safety restrictions, causing the CAFS measurements to have higher than average 950-hPa J-values. The latter can in part explain why the sign of the cloud effect is opposite between the simulations and CAFS for the Northern Pacific below 900 hPa. However, since a below-cloud enhancement has no physical basis in the current models, the difference in sign can also hint at possibly missing modeled cloud field types (Hall et al., 2018)."

**Comment 32:**

L308: Yes, but you know the Briegleb model is in error.

L313: The comparison with surface observations is a great addition.

L344: Wow, overall this comparison is great. It is good to see the difference between the real J (with albedo) and the observation with a black cloth. As a fair warning, this is not the correct way to compare with the observations. the light that is scattered up and then down, comes from a much wider region than can be blacked out from the black cloth under the instrument. When we tried such comparison with Fast-J (a long time ago), we had to do the calculation with an albedo of say 0.30 and then go into the RT solution and remove the direct upwelling (after the multiple scattering). Sorry, but that option is not currently available in Cloud-J.

**Reply 32:**

Thanks for sharing this insight. When we discussed this, it lead to the inclusion of line 340: "It is worth noting that this does not consider the portion of the upwelling radiation redirected into the downwelling field from atmospheric scattering, although this contribution is likely of secondary importance.".

**Comment 32:**

Fig 5 shows that EMEP-Ao (blue, the best attempt to match the observations) is indeed the best fit to the obs (orange). What is unusual here is that the EMEP-TB (which is clearly noted to have a surface albedo and thus overestimate J-O1D) is 30+% higher in all cases, and the EMEP-CJ (with the surface albedo) is only about 10% higher. If we go back to Fig.1, the TB Js are not that much higher in the BL for J-O1D. Can you explain?

**Reply 32:**

As noted in reply to comment 24, the latitude of the Lampedusa site (35.5N) falls within that of the N. Pacific ATom-1 block (20N-50N), where surface J-O1D rates from the TB scheme are greater than the

CJ rates. Given the large day-to-day variability in the CJ rates, it is also difficult to make out how the CJ and TB rates at the Lampedusa site 'should' compare to those over the ATom-1 N. Pacific block, though we think the TB rates being greater in Fig 5 is consistent with Fig 1c.

Comment 33:

L365: Very interesting, this daily variability with O3 column. Presumably it is the 100-200 hPa stratospheric column which at this location will vary because it is near the sub-tropical jet. That region keep shifting from stratospheric to tropospheric.

Figure 6: These results are very good, considering that MEGRIDOP O3 is monthly and that EMEP will be simulating the LMS (100-200 hPa) ozone via the boundary conditions at 100 hPa. Well done.

Reply 33:

Thanks. We agree that the day-to-day variability is likely the result of the O3 variations at the EMEP model top (the top layer where the BCs are imposed is also relatively thick).

Comment 34:

L395: Correct, it is seen in most of our Cloud-J tests. J-NO3 is most sensitive to clouds.

Fig.7: Looks like the J-H2O2 problem is with the corss sections being used?

L412 Fig.8: Wow, this looks so different from the other surface comparisons. Part of the problem here is that we are comparing new Js that we have not used before. Another part of the problem is that (I suspect) that all three surface obs. are not using the same cross sections and quantum yields to convert the irradiances to J-values.

L417: Good point about the O3 column – is there anyway to check it during the observations? However, the J-CH2O's both of them look pretty good.

L426: J-CH3COCHO (MGLY) is a very messy J-value, it has strong pressure dependent quantum yields and I doubt that TUV and the Chilbolton group are using the same formulation as Cloud-J.

Reply 34:

The surface observations indeed do not use consistent data sources for their quantum yields and cross section data.

Ozone column observations are in fact available for the Chilbolton site. Not including a more detailed analysis of the impact of the O3 column has simply been due to time limitations.

**Comment 34:**

L427: CJ has large visible wavelength bins, but the cross sections are averaged correctly at very high (<0.1 nm) resolution. So this is not the problem, look at the quantum yields. The acetone Js are another big problem with EMEP-Ao and EMEP-CJ greatly over predicting the 'observed' Js.

**Reply 34:**

Thanks for sharing this insight. We have now simply left out the statement regarding bin resolution.

**Comment 35:**

L432: What other optical properties? Clouds must dominate unless there was an haboob.

**Reply 35:**

Here variations in cloud optical properties broadly refer to variations in e.g. cloud thickness, altitude, and temperature, which are more factors determining cloud optical properties rather than optical properties of the clouds themselves. In the revised text, L432 has been rewritten as:

"The observed day-to-day variations in peak photolysis rate values can largely be attributed to daily variations in local cloud cover and other factors determining cloud optical properties, such as cloud thickness, altitude, and temperature (Walker et al., 2022)."

**Comment 36:**

Fig.9. OK, this figure is excellent. It shows that the corr. coeff. maxes out at ~0.90 for the best Js. It show the Js where we have clear problems with the physics being uses in CJ vs obs-J.

- J-O1d has consistent bias, the obs-J probably has old q-yields.
- J-H2O2and J-CH3CHO have similar problems
- J-CH2O (both) and J-NO2 look fine
- J-MGLY clearly is using different spectral data in the tow J calculations.

L442: This is a very important but surprising result. I am surprised that the correl.coeff. was not affected. Maybe it points out that the high-resolution cloud statistics have the same skill and RMSE as the low-resolution averages. There is a basic limit as to how well we can do cloud variability with the IFS (~ 0.9 corr. Coeff.)

L458: I think you mean R1 here. Also the Crutzen and Zimmerman paper is a classic, but terribly out of date if you want numbers. For current models (including yours?) the CO sink is 40-45%, not 70%.

**Reply 36:**

Thank you for sharing these insights, and for pointing out that R2 should indeed have been R1 (corrected). We have also updated the CO reference to Murray *et al.* (2021), changing the CO-sink number for the hydroxyl radical from 70% to 40%.

L459: Montzka 2011 only measure the variability in decay of CH3CCl3, the OH variability is inferred.

Reply 37:

Rephrased to 'as inferred by Motznka (2011)'.

Comment 38:

L470: Being part of the 'reference climatology', I can only say that it is not observed and is from an old model. It is valuable as a published 'reference' OH set, but should not be mistaken for ground truth.

L472-3: OK, so CJ gives higher than current model mean OH, but the J values match J-O1D surface observations (?) Fig 10, however, does single out the problem, the CH4 lifetime. Most of the CH4 loss is from the lower tropical atmosphere and CJ greatly increase OH there. Yet, we know that EMEP-TB is a poor estimate of J-O1D in the tropics, even though we may like the answers it gives in terms of OH.

L478: you cannot use 'overestimated' here since we do not know the correct value, only 'larger'

L479-480: It cannot be H2O since both EMEP models use the same H2O density- right? How much is the difference in trop O3 column between EMEP-CJ and EMEP-TB? You can also look at the ATom sensitivity of the CH4 and CO budgets to T, q, O3, NOx, CO, CH4, HOOH in Prather et al. 2023, Table 2, https://doi.org/10.5194/essd-15-3299-2023 Earth Syst. Sci. Data, 15, 3299–3349.

Reply 38:

We have rephrased 'overestimated' to 'comparatively high', since the lower tropospheric EMEP OH concentrations are on the higher end of the 17-model mean comparison.

Our reasoning behind including the H2O distribution in our list of possible sources of 'comparatively high' OH, is that the underestimated J-O1D values of EMEP-TB (as discussed earlier) may have masked the effects of high H2O concentrations. Though this of course purely speculative, and as noted in the text. The text has also been slightly rephrased to avoid speculations about water vapor or O3 concentrations possibly being 'excessive', given the difficulties in knowing what is right and wrong in terms of the OH concentrations.

Comment 39:

L482: Nice work on the aerosol photolytic effect with EMEP. I am surprised by the size of the change in the biomass burning regions.

L513: I cannot argue with the logic, but cannot see how CJ vs TB would underestimate the the surface O3 diurnal variations. Maybe you could compare the large-scale surface diel variations in O3 derived from many more site by Schnell et al Figure 1a-h (Atmos. Chem. Phys., 15, 10581–10596, 2015).

While the diurnal variations are not discussed in detail, Table 4 shows that the negative bias in daily mean O3 is less negative than that for O3max for EMEP-TB, indicating that diurnal variations are underestimated here as well (from experience, we know that the EMEP model often overestimates night-time O3 concentrations). The original manuscript text seemed to imply that CJ affects the diurnal cycle itself, while the EMEP-CJ results are in fact more consistent with CJ generally increasing surface O3 concentrations.

To highlight that CJ does not necessarily impact the diurnal variations themselves, the following sentence has been added after L513:

"However, the latter is also apparent in EMEP-TB, where the bias in daily mean O3 is less negative than that for O3max." Where "the latter" here refers to the diurnal variations in ozone.

Comment 40:

L513: OK, I have some problems with the estimate of computational costs (which can be found in the Cloud-J 2015 paper). So doing Briegleb every 60 min instead of 15 min we have 15% extra cost. You compare this with doing AvQCA every 15 min (250% cost). OK not a fair comparison. We only update J;s every hour, that should be fine. So, the AcQCA cost is now 60%. We know that the average extra calculations for AcQCA is2.8x, which would have the Briegleb being 20%. OK, close enough. So the choice is Briegleb at 15% or AvQCA at 45% (probably). I do not know how much the Table lookup costs, but the 15% is differential I presume. The 45% may be too much, but you should not that the vertical distribution of Js with Briegleb is the same as different cloud fields as you show. I think there will be some other changes in your budgets, but either way is a better representation of the clouds.

Reply 40:

Many thanks for critically evaluating these numbers. With the model time-step AvQCA scheme the EMEP run-time is increased by a factor of 2.5, while with the hourly Briegleb scheme the run-time is increased by a factor of 1.15. While the latter was correctly described as a 15% increase, the former was mistakenly written as a 250% rather than 150% increase. The text has now been corrected.

Dividing 150% by 3 to get a rough estimate of the hourly AvQCA increase (=50% total increase), and then dividing that by 2.8 to estimate the Briegleb cost (=18% total increase), makes much more sense compared to the 15% total increase found in the hourly Briegleb simulation.

Comment 41:

LL533 Fig 12: I am underwhelmed by this figure. Not sure what to make of the scattered results.

L544 Fig 13: Really fascinating daily comparison, actually quite impressive EMEP modeling with any J values!! Something wrong with Trinidad Head in the model. The EMEP model is doing an excellent job of following the maximum O3 levels. The shift in r and NMB from TB to CJ is noticeable, but the

bigger issue is the met fields and emissions. It still looks nice, but this plot is not the reason why one would choose CL over TB, I agree.

Reply 41:

The motivation behind Fig 12 stems from our long-standing problems with O3 inflow from the Atlantic sector. In the regional (UNECE excl. NA) EMEP configuration, O3 boundary conditions are typically adjusted by applying a monthly mean observed 'Mace Head correction factor', effectively reducing the inflow of O3. Not having to apply this correction factor would be much preferred, but for that we would first need to better understand what drives the simulated inflow of O3. Fig. 12 therefore shows information that might be useful to us in later work, highlighting that CJ further adds to the Atlantic inflow of O3.

Comment 43:

Final note. You have taken the Cloud-J v7.3e which is a solid version of Cloud-J and includes all the cloud overlap options. There has been further development of Cloud-J since 2015 and you and others should look at what has been fixed or updated (like physically based ocean surface albedos). I include the notes of the successive changes below. In general, I recommend you adopt Cloud-J v8.0c (which is being implemented in GEOS-Chem now) is you are only interested in J-values but v7.6 if you want the option of calculating solar heating rates in addition to J-values.

Reply 43:

Thank you for these recommendations, implementing Cloud-J v8.0c seems to be the logical next step.

**References**

Murray, L. T., Fiore, A. M., Shindell, D. T., Naik, V., & Horowitz, L. W. (2021). Large uncertainties in global hydroxyl projections tied to fate of reactive nitrogen and carbon. *Proceedings of the National Academy of Sciences, 118*(43), e2115204118.

**Response to RC2 (anonymous)**

Comment 1:

Line 67: the reference for ECLIPSEv6b should be given

Reply 1:

Reference has been added to the International Institute for Applied Systems Analysis (IIASA) webpage that describes the ECLIPSEv6b data set.

Comment 2:

Line 76: what definition of all(cloudy)/clear sky do you use ? It should be precised, since the distinction is used throughout the article.

Reply 2:

In the revised manuscript, the text has been changed to highlight that clear-sky refers to an atmosphere without clouds and aerosols, cloudy-sky to an atmosphere with clouds but no aerosols, and all-sky to an atmosphere which may contain both clouds and aerosols.

Comment 3:

Lines 78-79 Could you give the cloud optical depth (for instance at 550 nm ) ?

Reply 3:

Unfortunately we no longer have access to the original code or input data that was used to create the tabulated photolysis rates about 25 years ago, and so this information is lost in time.

Comment 4:

Line 95-96: does the use of more streams changes the results ? Since one of the differences between Cloud-J and EMEP-TB lies in the number of streams, it would be interesting to know.

Reply 4:

While the number of streams undoubtedly has an impact on the accuracy of the results, we do not have the capacity to run dedicated Cloud-J experiments with different numbers of streams. However, the cited Wild et al. (2000) paper includes a discussion on the impacts of choosing two-stream over eight-stream methods.

Line 96: could you precise whether it uses discrete ordinates method or not ?

Reply 5:

Cloud-J (with predecessor Fast-J) solves the full equations of radiative transfer at 4 upward and 4 downward Gaussian quadrature points (i.e. discrete ordinates; see Appendix A from Wild et al. (2000)). This information is now added to the manuscript text, which is updated to read:

"Its highly optimized eight-stream radiative transfer scheme, solving the specific intensity of the radiation field at four upward and downward Gaussian quadrature points, employs 18 wavelength bins for wavelengths between 177 nm and 778 nm (Neu et al., 2007), spanning the wavelengths relevant to tropospheric and stratospheric chemistry."

Comment 6:

Lines 95-100: do you use methods such as delta-M in the presence of highly anisotropic phase functions ? If no, have you made trials to show that is does not bring significant improvements ?

Reply 6:

As described in Wild et al. (2000) for the Fast-J predecessor code, the Cloud-J algorithm employs the Legendre expansion of the exact scattering phase functions, requiring no adjustment to optical depth (delta-M) or extinction coefficients.

Comment 7:

Lines 95-100: do you use methods to correct the intensity, for instance the one of Nakajima and Tanaka (JQSRT 1988) ?

Reply 7:

The Fast-J/Cloud-J radiative transfer solution requires no correction to the intensity, as described in Wild et al. (2000).

Comment 8:

Lines 111-115 a basic description (without dwelling into details) on MAX-COR AvQCA and the Briegleb method would be interesting.

Reply 8:

To make the text regarding the MAX-COR AvQCA and Briegleb schemes more clear, the short description of the Briegleb scheme (lines 237-238 of the original manuscript) has been moved to Section 2.3, where the Briegleb scheme is introduced in the original manuscript. For the MAX-COR AvQCA scheme, we have also added the information that the average 2.8 photolysis rate calculation calls per ICA, depends on the ranges of optical depths present in the (vertical) grid-cells of the ICA. We note that a detailed description of the MAX-COR AvQCA scheme is given in the cited work of Prather (2015).

**Comment 9:**

Lines 153-154 could you precise what do you mean by a "climatology based on these years" ? Do you extrapolate after 2021 ? If so, how ?

**Reply 9:**

In response to RC1, Section 2.3.2 has been considerably rewritten. This included the rephrasing of "climatology based on these years" to "multi-year monthly mean climatology based on these years", with 'these years' in the text referring to 2002-2021. As noted in the text, this monthly mean climatology is applied for simulation years both before 2002 and after 2021.

**Comment 10:**

Paragraph 2.3: you do not mention what TOA (Top Of Atmosphere) SSI (Solar Spectral Irradiance) you are using. Have you tried more than one to asses their influence on the photolysis rates computed by Cloud-J ? The SSI seems to me an important part of a model which computes photolysis rates.

**Reply 10:**

We have now added that Cloud-J employs a SSI based on irradiance measurements from the Solar Ultraviolet Spectral Irradiance Monitor (SUSIM) instrument onboard the Upper Atmosphere Research Satellite (UARS), averaged between conditions representative of solar minimum and solar maximum.

While we would expect that variations in TOA SSI are perhaps most relevant to stratospheric photochemistry, it would be interesting to see to what extent tropospheric J-values are affected. Though a detailed study if the impact of the TOA SSI is beyond the scope of the current work.

**Comment 11:**

Lines 225-226: was the conjugate dataset constructed from TUV alone or also from CAFS ?

**Reply 11:**

To our knowledge, no CAFS actinic flux measurements were used to construct the TUV-based conjugate clear-sky data set. Details of the methodology behind the construction of the conjugate data set are described in the cited work of Hall et al. (2018), which we repeat here in brief:

The TUV model was used to generate actinic fluxes along the ATom-1 flight paths, using an eight-stream discrete ordinate radiative transfer method with a pseudo-spherical modification, no clouds and no aerosols, a fixed surface albedo of 0.06, and ozone column data from the Ozone Monitoring Instrument (Hall et al, 2018). These TUV-generated fluxes were then processed using the same photolysis frequency code to ensure consistency between the quantum yield data, cross-section data, and pressure and temperature dependencies.

Line 250 : Could you define what the WASM scheme is ?

Reply 12:

We have now added the definition "WRF-single-moment-microphysics class 5 (WSM5)", noting that the A in WASM was a typo in the original manuscript.

Comment 13:

Line 254: what is the tropopause altitude ? Please specify or plot a vertical temperature profile.

Reply 13:

Since the exact tropopause height is not strictly relevant to the interpretation of the data shown in Figure 1, we have rephrased "tropopause" to "upper troposphere", to indicate that it refers to the higher altitude regions shown in this figure.

Comment 14:

Line 261: difference with what quantity ?

Reply 14:

We have changed "differences" to "differences for these setups", to better reflect that they refer to the J-values simulated by the IFS-based Cloud-J model configurations.

Comment 15:

Line 261-262: What do you mean by "comparatively largest differences" ? I can see some discrepancies between the text and the figures

1. for J-NO2 according to figure 2, CJB is closest from CAFS than CJVL above 800 hPa for both zones, in the text you say that it presents the largest differences for both zones.

2. for J-O1D accord to figure 2, for tropical pacific, CJ15 is closest from CAFS than other cloud-J simulations, in the text you say that it presents the largest differences for both zones.

3. You also say that, for J-NO2, the largest differences occur for CJB for both zones (line 261), and for CJVL in the the Northern Pacific zone. This is not consistent.

Reply 15:

1. We have clarified in the text that the 'differences' refer to the differences between the different IFS-based Cloud-J model configurations (response to comment 14), and not relative to the CAFS measurements.

2. As point 1.

3. Thanks for pointing out this inconsistency. For the IFS-based Cloud-J J-NO2 values in Northern Pacific block, in fact both CJVL and CJB show similar (but opposite in sign) variations to the values calculated by CJ and CJ15.

In the revised manuscript, the description of these 3 points has been clarified. Furthermore, the revision revealed that the statement that "Both the results for CJVL and CJB are consistent with the Northern pacific being generally more cloudy, and thus experiencing a larger impact of the representation of the cloud effect" is not logical, since CJB also shows differences with respect to CJ in the Tropical Pacific block. This statement has therefore been removed in the revised manuscript.

Comment 16:

Line 271: the altitude of the tropopause would be useful.

Reply 16:

Since the upper and middle troposphere referenced on L271 does not have to be strictly defined in the context of Figure 3, "upper and middle troposphere" has been rephrased to "above ~800 hPa" to more closely correspond to what is shown in Figure 3.

Comment 17:

Line 291: "diminished" => "diminished by more than 2.5 %"

Line 291: "enhanced" => "enhanced by more than 2.5 %"

Reply 17:

Done.

Comment 18:

Line 294: the crosses are not always located on the thick part of the lines.

Reply 18:

Reference to the crosses being placed on this line segment has simply been left out in the revised manuscript, since they are indeed located on the thick parts.

Comment 19:

Lines 289-298: for figure 4, what determines the horizontal position of the bars ? The legend of the horizontal axe should be precised. Their length is always 100 %, but they do not have the same abscissas. Detailing an example would help the reader.

Reply 19:

The thin bar segment describing the number of percentage points (of J-value measurements) that were neither increased or decreased by more than 2.5% is always centered on zero on the x-axis. The length of the thick bar-segment to the right of the thin line spans the number of percentage points for which J-

values were increased by more than 2.5%, whereas the length of the thick bar-segment to the left of the thin line spans the number of percentage points for which J-values were reduced by more than 2.5%.

To clarify this, we have added ", with the thin line being centered on zero on the x-axis." to L294 of the original manuscript. In addition, the example started on line 296-298 describing the horizontal bar for the CAFS J-O1D measurements between 100-300 hPa, has been extended to describe the full horizontal bar.

Comment 20:

Line 320-329: could you give more details about the cloud cover ?

Reply 20:

In the description of the ChArMEx campaign, we have added "Furthermore, monthly mean cloud cover values for June 2013 were below 40% along the Mediterranean coast near to Lampedusa, with the full time-series of observed photolysis rates also showing little evidence of cloud-effects (Fig. S2)."

Comment 21:

Line 326: "The moderate aerosol mass... by marine aerosols". I think you meant that, in mass, the main aerosol constituent was marine aerosols.

Reply 21:

We have rephrased the sentence to exclude the statement about the contribution of marine aerosol. As is evident from the cited text of Mallet et al. 2016, marine aerosol undoubtedly made a large contribution to the total aerosol mass (for example, based on their choice of value of 0.75 for the factor s to correct the mass concentration of sea-spray aerosols only). However, exact numbers of the contribution of marine aerosol to the total aerosol mass are not given, and therefore we should have been more careful with our statement.

Comment 23:

Line 339-343 is it possible for Cloud-J to distinguish between the downward actinic flux and the upward one ? it would allow to take into account the portion of the upwelling flux redirected downwards. What was the albedo of the surface (not in the model !), if close to zero, it would avoid the problem.

Reply 23:

Paraphrasing RC1 comment 32: Removing the upwelling flux redirected downwards (after the multiple scattering) is not currently an option in Cloud-J.

The experimental setup of the measurements described on lines 339-343 included a matt black shadowing ring to limit the upwelling flux. However, as noted by RC1, this is only an approximate zero-albedo configuration, since upwelling flux redirected downward can come from quite a wide region.

Comment 24:

Lines 342-343 I understand that the photolysis rates used in the chemistry were computed using a non-zero albedo. If I'm right, it should be explicitly stated.

Reply 24:

The photolysis rates calculated with zero albedo are indeed only used for model output, while the photolysis rates used in the actual chemistry include a non-zero surface albedo (to not change the chemistry calculations between the non-zero and zero albedo numerical experiments). To highlight this distinction, "whereas the photolysis rates used in the chemistry remain unchanged."(line 343) has been changed to "whereas the photolysis rates used in the chemistry remain unchanged (i.e., non-zero albedo)".

Comment 25:

p18: figure 5 could be separated in two figures, one with the photolysis rates, the other with the ratios model_rate/observed_rate.

Reply 25:

While we agree that such a figure would be useful, the ratio's are implicitly discussed in the text for the hourly noon-time J-values, discussing for example their %-differences in some detail. We therefore think that adding a figure showing the model_rate/observed_rate would not necessarily add a layer of depth to the analysis that is currently missing.

Comment 26:

p19, figure 4: could you add a second vertical scale on the right to plot the simulated column below 100 hPa ?

Reply 26:

The contribution of the MEGRIDOP satellite data amounts to 247 DU throughout the time period shown in Figure 6. The simulated column below 100 hPa can therefore be constructed by subtracting 247 from the vertical scale. To clarify this point without the addition of a second vertical scale, line 372 of the original manuscript has been rephrased, to stress that the above-100 hPa O3 amounted to 247 DU throughout the time period shown in Figure 6.

Comment 27:

Line 395: "the impact of clouds ... for both model and observation" Where do we see it ? Were some days cloudy ?

Reply 27:

Our statement that the prevailing meteorological conditions were clear-sky was imprecise, now being rephrased to 'cloud-free with moderate aerosol loading'.

Nevertheless, the variations in EMEP-TB for J-NO3 reveal that at least some cloud cover was present in the model (since cloud cover is the only factor affecting the day-to-day variability of the tabulated photolysis rates). Acknowledging that our original statement was imprecise, because it excluded the possibility of aerosols having an effect, line 395 has been rephrased as:

"For the latter, the impact of any (light) cloud or aerosol presence is comparatively the largest both for model and observation, possibly owing to its almost exclusive dependence on wavelengths in the visible spectrum."

Comment 28:

p21, Figure 7, plotting also the ratio simulated/observed would be interesting and allow a more easy comparison between the models.

Reply 28:

Since the intention of Section 4.3 is not to give a detailed analysis of each of the 8 shown J-values, but rather to give a general impression of the model skill and inter-model differences, we think that adding figures showing the ratio simulated/observed would not necessarily add a lot of depth, thereby deciding not to make changes to Figure 7.

Comment 29:

p32, figure 9: could you provide a parameter to assess the quality of the fit, e.g. mean squared residual ?

Reply 29:

The r-squared value is a global measure of the variance explained by the linear fit, while we show only the r-values (linear correlation coefficients; or Pearson correlation coefficients). Since the r-values shown in the plots can easily be squared, we think figure 9 provides adequate information about the quality of the fit. For example, taking the worst-case EMEP-A0 result (r = 0.81 for J-CH3COCHO, panel i), the linear fit explains 65% (0.81*0.81=0.65) of the variance. Since the figure is principally about the EMEP-A0 comparison, our conclusion is that the linear fits are adequate for the model-to-measurement comparison, even though linear relationships between the modeled and measured hourly noon-time values are not necessarily expected.

Comment 30:

Line 463: Over which period was the reference climatology from Spivakovsky computed ? The intercomparison of Naik done ?

Reply 30:

From the Spivakovsky et al. (2000) paper, their work appears to combine a number of independent climatological data sets (cloud cover, temperature, atmospheric constituents such as O3, etc.) to

calculate a single climatological OH distribution. Since, for example, the temperature climatology is based on monthly averages between 1986-1989, while the O3 climatology is in part based on measurements between 1978-1992, no single range of years can be associated with the resulting OH climatology.

The simulations for the model-intercomparison data from Naik et al. (2016) were performed for the year 2000, which has now been added as additional information to the revised manuscript text.

Comment 31:

Line 472-477. Two references are used to asses the models : the Spivakovsky climatology, and the Naik intercomparison. Which one is the best ?

Reply 31:

These two references are included to see how the EMEP model configurations compare to earlier published literature, without necessarily trying to determine which EMEP setup or reference model is best. Knowing which OH distribution is closest to reality is not an obvious thing, considering also the challenges in deriving OH concentrations from observations (of other trace gases).

Comment 32:

p26, figure 10: could this figure be replaced or completed by a table with the values ?

Reply 32:

This is a good point, as having the exact values available from literature has helped us a lot in our work. We have now added a table to the supplementary material showing the numbers that were used to construct Figure 10.

Comment 33:

p26, section 5.2 and p27, figure 11 It would be interesting to run simulations taking into account the radiative contribution of only one type of aerosol (dust, sea salt, biomass burning aerosols), and to plot the results in figures similar to figure 11.

Reply 33:

While simulations for each individual aerosol species would be interesting, our impression is that the results presented in Figure 11 are clear enough to distinguish the effects of the individual aerosol species, at least in the context of the (global) analysis presented in this work. That is, the individual impacts of sea salt, biomass burning, and dust are not necessarily obscured by considering the photolytic impacts of these aerosol species all at once, given that each species has at least one clearly identifiable source region (oceans, tropical biomass burning regions, and deserts, respectively).

Line 509-510: I think that "decrease" in "increase" mean decrease/increase from the TB values. Could you precise it ?

Reply 34:

In the revised manuscript, we have clarified that this increase/decrease is indeed relative to EMEP-TB.

Comment 35:

Line 521: the decrease of performance does not seem obvious for the United Kingdom.

Reply 35:

We agree that this is not obvious for the United Kingdom (UK) as a whole, while also realizing that mentioning only the UK is not very accurate here (given that Ireland is not a part of the UK). We have therefore replaced the UK with "western and northern stations of the United Kingdom and Ireland" in the revised manuscript.

Upon closer inspection the stations on the Iberian Peninsula do in fact all lie in Spain, with no stations being located in Portugal. The text has been updated to reflect this, changing "Western coast of Spain" to "Western Spain", and omitting Portugal from the list.

Taken together, "the performance is decreased along the Western coast of Spain, Portugal, and the United Kingdom" has been  changed to "the performance is decreased in Western Spain, and the western and northern stations of the United Kingdom and Ireland".

Comment 36:

A map showing the positions of the four sites used would be useful.

Reply 36:

In the revised manuscript, we hope to have addressed this comment by including the country or region in which each of the stations is located in the text, as well as in the plot titles.

Comment 37:

Fig 13: It would be more clear, to use, for each site, one vertical scale for the plot obs = f(day of year), and another for the ratios EMEP-CJ/obs and EMEP-TB/obs. The information about the variation over a year would be conserved, and the ratio simulation/observation would be more easy to see.

Reply 37:

Our intention with Figure 13 is to illustrate how the EMEP model configurations perform over the course of a year, without going in to detail about where and when exactly the models perform well or poorly. To this end, the plot-style of Figure 13 follows that of Stadtler et al. (2018) (Figures 6, 7, and 8). While we agree that showing the ratio's between EMEP-CJ/obs and EMEP-TB/obs would provide additional information, e.g. by more clearly highlighting when EMEP-CJ differs the most of EMEP-

TB, we also think that adding this information goes beyond the message that we are trying to convey in this section (5.3.1).

Comment 38:

In fig 13, we see that the absolute values of the NMB is higher for CJ than for TB. This could be understood as a decrease of the performances of CJ relatively to TB. It should be mentioned in the text and explained.

Reply 38:

In the text we mention that the EMEP-CJ NMB bias is worsened for the Trinidad Head and Tudor Hill stations (upwind and downwind of the United States), while also mentioning that the absolute NMB for the Mace Head and Ryori stations is comparable to that of EMEP-TB. With this, we hope to have made a fair assessment of the change in (NMB) performance of the EMEP-CJ model relative to EMEP-TB.

Comment 39:

Line 551-552: the "the efficacy ... from the CAFS instrument". This is true for clear sky, but less for cloud sky, according to figure 2.

Reply 39:

We agree that Figure 2 illustrates that the use of Cloud-J by no means implies that the simulated cloudy-sky photolysis rates are perfect compared to observation. Our interpretation is nevertheless still that Cloud-J performs favorably and robustly against observations, given the range of numerical experiments discussed in this work. To change the tone on lines 551-552, 'efficacy' has been replaced with 'performance'.

Comment 40:

Line 557: (sic. 567)

1. The general improvement is not obvious for global sites, see remark for paragraph 5.3.1., it should be precised.

2. For Europe, the number of site for which CJ improves the performance is greater than the number of sites for which it is decreased. "General improvement" is not the best way to sum it up.

Reply 40:

1. As mentioned in reply to comment 38, our interpretation is that the NMB is only really worsened at the Trinidad Head and Tudor Hill stations, for which the NMB was positive already in EMEP-TB. The correlation coefficient improved considerably at the Trinidad Head station (from 0.58 to 0.64), and from 0.84 to 0.86 at the Tudor Hill station. Given that the correlation coefficient also increases at the Ryori station (but stays the same at Mace Head), our interpretation of Figure 13 is nevertheless that it illustrates a general improvement of EMEP-CJ over EMEP-TB.

2. The wording has been changed to better reflect that the average values for O3max shown in Table 2 show an improvement, while now also highlighting that this is relative to the observed concentrations.

Comment 41:

The description of the role of the different files should be more detailed, to help the reader understand the global structure of the model.

Reply 41:

We hope to have addressed this comment by including reference (DOI and github page (https://github.com/metno/emep-ctm)) to the latest EMEP MSC-W opensource release (October 2023) in the "model and data availability" section, which includes the Cloud-J upgrade. In addition, we have included reference to a webpage that gives additional EMEP model background information: https://emep-ctm.readthedocs.io/en/latest/

Comment 42:

In the python scripts:

- it would be useful to have comment to link the plot commands to the figures of the article.

- some paths should be defined as parameters, for instance, /home/willemvc/Desktop does not exist everywhere, the user should be able to change once and for all these paths (in a python module, for instance).

Reply 42:

We agree that it would be best to have the python scripts (or commands therein) refer to the plots used in the manuscript. However, since the names of the scripts are intentionally descriptive (e.g., "ATom.py" can be used to produce the figures showing ATom-1 comparisons, and "global_OH.py" to create Figure 10), we would prefer not to create a new ZENODO upload just for this.

It almost seems inevitable that users who would want to use the scripts will have to change the paths to the input files, with the variable controlling the paths clearly being noted in each of the scripts (e.g., "path", or "base_f" for base folder). Currently, it is also beyond our capacity to set the scripts up to run within a custom Python module.

Comment 43:

Figure 1, a and b "Tr" => "Trop" or "Tropical", it would be more clear and is enough space for it.

Figure 1, c and d, "N." => "North", for the same reason.

Line 519 : "absolute annual mean" => "absolute value of the mean"

Reply 43:

"Tr" => "Trop" and "N." => "North" has been changed in Figures 1, 2, and 3.

"absolute annual mean" => "absolute value of the mean" has also been changed.

Lines 555 - 559 : "the Cloud-J based simulations show ... below 350 nm" Could you recall the sections of your article were these assertions are detailed ?

Reply 43

Since these statements relate to certain specific model-to-measurement comparisons, we agree that it is useful to refer to the sections supporting the specific conclusions. Lines 555-559 now read:

"While the general performance of Cloud-J is encouraging, the Cloud-J based simulations show a tendency to overpredict the frequency of occurrence and magnitude of above-cloud enhancements and below-cloud diminishments (Section 3), while also underestimating wintertime photolysis rates sensitive to wavelengths below 350 nm (Section 4.3)."

**References**

Hall, S. R., Ullmann, K., Prather, M. J., Flynn, C. M., Murray, L. T., Fiore, A. M., ... & Archibald, A. T. (2018). Cloud impacts on photochemistry: building a climatology of photolysis rates from the Atmospheric Tomography mission. Atmospheric Chemistry and Physics, 18(22), 16809-16828.

Wild, O., Zhu, X. I. N., & Prather, M. J. (2000). Fast-J: Accurate simulation of in-and below-cloud photolysis in tropospheric chemical models. Journal of Atmospheric Chemistry, 37, 245-282.

Prather, M. J. (2015). Photolysis rates in correlated overlapping cloud fields: Cloud-J 7.3 c. Geoscientific Model Development, 8(8), 2587-2595.

Stadtler, S., Simpson, D., Schröder, S., Taraborrelli, D., Bott, A., & Schultz, M. (2018). Ozone impacts of gas–aerosol uptake in global chemistry transport models. *Atmospheric chemistry and physics*, *18*(5), 3147-3171.